# γ-ORTHOGONALIZED TENSOR DEFLATION: TOWARDS ROBUST & INTERPRETABLE TENSOR DECOMPOSITION IN THE PRESENCE OF CORRELATED COMPONENTS

## ABSTRACT

We tackle the problem of recovering a low-rank tensor signal with possibly correlated components from a random noisy tensor, or the so-called *spiked tensor model*. When the underlying components are orthogonal, they can be recovered efficiently using *tensor deflation*, while correlated components may alter the tensor deflation mechanism, thereby preventing efficient recovery. Relying on recently developed tools from random tensor theory, we deal precisely with the non-orthogonal case by deriving an asymptotic analysis of a *parameterized* deflation procedure, which we refer to as γ-orthogonalized tensor deflation. Based on this analysis, an efficient tensor deflation algorithm is proposed by optimizing the parameter injected into the deflation mechanism, which in turn is proven to be optimal by construction for the studied tensor model. We perform a detailed theoretical and algorithmic analysis on the rank-2 order-3 model, and outline a general structure to tackle the problem in more generality for arbitrary ranks/orders, aiming to lead to a broader impact in machine learning and beyond.

## 1 INTRODUCTION

Tensor methods have been proven to be a powerful and versatile tool to model multi-relational, multimodal and/or temporal data (Jenatton et al., 2012; Nickel et al., 2011; Rabanser et al., 2017; Sidiropoulos et al., 2017; Fawzi et al., 2022), all of which lie at the heart of the current state-of-the-art in machine learning research. In the era of exascale models, one of the most important fundamental problems is extracting latent structure from high-dimensional data, which hints at more sophisticated foundational abilities learned by the model, showcasing its ability to meta-learn. Typically, such latent structure is of low rank and includes correlations among the different signal components. Tensor methods, as in (Anandkumar et al., 2014; Zare et al., 2018), have been quite successful in tasks of unsupervised nature, among others.

In particular, we shed light on the spiked tensor model. A rank-r order-d spiked tensor model is written as follows

$$\mathcal{T} \equiv \sum_{i=1}^{r} \bigotimes_{j=1}^{d} \beta_i x_j^{(i)} + \frac{1}{\sqrt{n}} \mathcal{W} \in \mathbb{R}^{p_1 \times p_2 \times \ldots \times p_d},$$

where $\beta_i \geq 0$ correspond to the signal-to-noise strengths and $x_j^{(i)}$ are unit vectors.

In practice, this model corresponds to a variety of applications, across diverse domains. In compressed sensing (CS) (Lim & Comon, 2010; Sidiropoulos & Kyrillidis, 2012; Yang et al., 2015), it is used to model images that are sparse in some dictionary or lying in a union of lower dimensional subspaces, for instance natural images or medical images (undersampled MRI, CT etc). In reinforcement learning (RL) (Rozada & Marques, 2022; Agarwal et al., 2020; Ni et al., 2021), it is used to model the transition kernel over a very large state space, yet exhibiting a low rank structure. In natural language processing (NLP) (Agarwal et al., 2020; Anandkumar et al., 2013; Cheng et al., 2015), orthogonalized deflation based tensor decomposition (Mackey, 2008) and its variants are, for instance, used for topic modeling.

**Related Work.** We consider a more detailed literature review in Appendix A, given the space constraints.

**Notation.** The set $\{1, \ldots, n\}$ is denoted by $[n]$. The unit sphere in $\mathbb{R}^p$ is denoted by $\mathbb{S}^{p-1}$. The Dirac measure at some real value $x$ is denoted by $\delta_x$. The support of a measure $\mu$ is denoted by $\text{supp}(\mu)$.

The inner product between two vectors $u, v$ is denoted by $\langle u, v \rangle = \sum_i u_i v_i$. The imaginary part of a complex number $z$ is denoted by $\Im[z]$. The set of eigenvalues of a matrix $\boldsymbol{M}$ is denoted by $\mathrm{Sp}(\boldsymbol{M})$. The almost sure converges is denoted by the arrow $\xrightarrow{\text{a.s.}}$. The notation $a_n \asymp b_n$ means that $a_n$ and $b_n$ converge to the same limit as $n \to \infty$.

**Preliminaries.** We provide a self-contained review of some of the key concepts of random matrix/tensor theory (RMT/RTT) to guide the reader throughout the theoretical results of the paper. We outline that in Appendix B.

**Key Contributions.** Our contributions are outlined as follows

- We introduce $\gamma$-orthogonalized tensor deflation, a variant of the orthogonalized tensor deflation algorithm by injecting an optimizable parameter $\gamma \in [0, 1]$ into the projection hyperplane, as described in §3. In particular, $\gamma = 1$ corresponds to the classical orthogonalized deflation (Mackey, 2008) and $\gamma = 0$ corresponds to projecting onto the signal subspace itself.

- We carry out a random tensor theory (RTT) analysis of the proposed deflation method applied to a *noisy rank-two asymmetric spiked tensor model*[1], as defined in §2.1.

- Relying on the theoretical construction performed throughout the paper, we design an efficient tensor deflation algorithm (§3) that compares favorably (as measured by alignments) to state-of-the-art and is fully theoretically tractable with the developed theory, as compared to some of its learning-based counterparts.

## 2 LOW-RANK SPIKED TENSOR MODEL

We study the problem of recovering a low-rank $r$ (possibly high-order $d$) signal from a noisy full rank tensor, which formulates as follows

$$\mathcal{T} \equiv \sum_{i=1}^{r} \bigotimes_{j=1}^{d} \beta_i x_j^{(i)} + \frac{1}{\sqrt{n}} \mathcal{W} \in \mathbb{R}^{p_1 \times p_2 \times \dots \times p_d},$$

where $\beta_i \geq 0$ correspond to the signal-to-noise ratios (SNRs), $x_j^{(i)} \in \mathbb{S}^{p_j - 1}$ with $\|x_j^{(i)}\| = 1$, $\forall\, i, j$ are the signal components. $\mathcal{W}$ is a random (possibly asymmetric) tensor with standard Gaussian i.i.d. entries, i.e., $W_{ijk} \sim \mathcal{N}(0, 1)$, and $n = \sum_{i=1}^{d} p_i$.

**Correlated Signal Components.** When the signal components are orthogonal $\left( \langle x_j^{(i_1)}, x_j^{(i_2)} \rangle = 0,\ i_1 \neq i_2 \right)$, a tensor deflation algorithm step would recover the signal component with the highest SNR. In a variety of practical scenarios, this assumption is quite restrictive. On a further note, tensor decomposition algorithms that rely on this assumption perform quite poorly when the latter is not satisfied, as shown in Figure 1. We alleviate this assumption by allowing arbitrary correlations among the signal components.

**Alignments as an Evaluation Metric.** Since the signal components are normalized, our goal is to maximize the alignments of the recovered signal with the hidden low-rank signal across modes. Indeed, in high dimension, the probability that two random vectors $u, v$ are orthogonal, i.e. $\langle u, v \rangle = 0$, is very high, which shows the difficulty of obtaining high estimation accuracies (alignments) in this setting.

### 2.1 RANK-2 ORDER-3 SPIKED TENSOR MODEL

Given the simplicity to visualize it and the focus on the inter-component correlation, we consider the following rank-2 order-3 spiked tensor model

$$\mathcal{T}_1 \equiv \sum_{i=1}^{2} \beta_i x_i \otimes y_i \otimes z_i + \frac{1}{\sqrt{n}} \mathcal{W} \in \mathbb{R}^{p \times p \times p}, \tag{1}$$

with $n = 3p$ in this case.

---

[1]Spiked models are more both general and offer numerous theoretical advantages compared to unrealistic noiseless low-rank models. For instance, subtracting a best rank-1 approximation of a noiseless low-rank tensor may increase its rank (Stegeman & Comon, 2010), while spiked models do not suffer from such limitation since their noise component is full rank (Strassen, 1983).

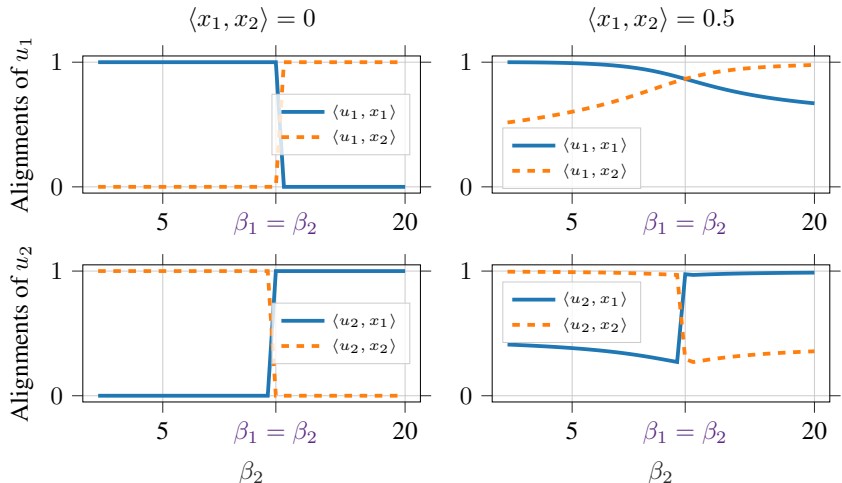

Figure 1: **Orthogonalized deflation** (as defined in §3) is applied to the noiseless rank-two tensor model $\sum_{i=1}^{2} \beta_i x_i^{\otimes 3}$, where $x_i \in \mathbb{R}^{200}$, $\beta_1$ is fixed and $\beta_2$ varies along the $x$-axis. $u_1$ and $u_2$ are the resulting signal estimates after one and two steps of orthogonalized deflation respectively. **Left:** $u_1$ and $u_2$ align well with the hidden signal $x_1$ and $x_2$ in the orthogonal case, i.e $\langle x_1, x_2 \rangle = 0$, highlighting a high estimation quality. **Right:** In the presence of correlation among the signal components, i.e $\langle x_1, x_2 \rangle = 0.5$, $u_1$ and $u_2$ don't align well with the hidden signal components $x_1$ and $x_2$, highlighting a poor signal recovery quality.

**Assumption for Mathematical Convenience.** In this setting, we further assume that the alignments among signal components are uniform across the modes, i.e.

$$\alpha \equiv \langle x_1, x_2 \rangle = \langle y_1, y_2 \rangle = \langle z_1, z_2 \rangle. \tag{2}$$

We highlight here that $\alpha$ measures the degree of correlation among the signal components. The choice of a uniform $\alpha$ across the signal modes equation 2 is for mathematical convenience only. Extending to general alignments $\alpha_x$, $\alpha_y$, $\alpha_z$ is rather straightforward, but would probably require the use of a solver, SymPy (Meurer et al., 2017) for instance, as the theoretical alignment expressions become quite tedious.

## 3  $\gamma$-ORTHOGONALIZED TENSOR DEFLATION

We present a strategy to recover the hidden low-rank signal components from a noisy rank-2 order-3 spiked tensor model, while jointly estimating all model parameters.

### 3.1  CONSTRUCTION OF THE DEFLATION MECHANISM: RANK-2 ORDER-3 SPIKED TENSOR MODEL

**First Deflation Step.** This step consists of a best rank-1 approximation of $\mathcal{T}_1$ (1), which is written as $\hat{\lambda}_1 \hat{u}_1 \otimes \hat{v}_1 \otimes \hat{w}_1$.

**Second Deflation Step.** Now, given $\hat{u}_1 \in \mathbb{S}^{p-1}$, we perform a best rank-1 approximation $\hat{\lambda}_2 \hat{u}_2 \otimes \hat{v}_2 \otimes \hat{w}_2$ of the resulting tensor from the first deflation step

$$\mathcal{T}_2 \equiv \mathcal{T}_1 \times_1 \left( \boldsymbol{I}_p - \gamma \hat{u}_1 \hat{u}_1^\top \right) = \mathcal{T}_1 - \gamma \hat{u}_1 \otimes \mathcal{T}_1(\hat{u}_1), \tag{3}$$

for some tunable parameter $\gamma \in [0, 1]$. If $\gamma = 1$, this step reduces to performing the classical *orthogonalized deflation* (Mackey, 2008).

**Best Rank-1 Approximation conditions.** Best rank-1 approximations from first and second deflation respectively satisfy the following identities, for $i \in [2]$

$$\mathcal{T}_i(\cdot, \hat{v}_i, \hat{w}_i) = \hat{\lambda}_i \hat{u}_i, \quad \mathcal{T}_i(\hat{u}_i, \cdot, \hat{w}_i) = \hat{\lambda}_i \hat{v}_i, \quad \mathcal{T}_i(\hat{u}_i, \hat{v}_i, \cdot) = \hat{\lambda}_i \hat{w}_i, \quad \hat{\lambda}_i = \mathcal{T}_i(\hat{u}_i, \hat{v}_i, \hat{w}_i). \tag{4}$$

**Tensor Power Iteration (TPI).** Initialized with SVD (Auddy & Yuan, 2022), TPI allows us to compute $\hat{\lambda}_1 \hat{u}_1 \otimes \hat{v}_1 \otimes \hat{w}_1$ and $\hat{\lambda}_2 \hat{u}_2 \otimes \hat{v}_2 \otimes \hat{w}_2$ in practice. For the orthogonal case ($\alpha = 0$), it has been proven to converge in polynomial time for $\beta_i \geq O(p^{3/2})$. Now, for each deflation step $i \in [2]$, we denote the following alignments as

$$
\begin{aligned}
\hat{\rho}_{1i} &\equiv |\langle \hat{u}_1, x_i \rangle| \asymp |\langle \hat{v}_1, y_i \rangle| \asymp |\langle \hat{w}_1, z_i \rangle|, \quad \hat{\theta}_{2i} \equiv |\langle \hat{u}_2, x_i \rangle|, \\
\hat{\rho}_{2i} &\equiv |\langle \hat{v}_2, y_i \rangle| \asymp |\langle \hat{w}_2, z_i \rangle|, \quad \hat{\kappa} \equiv |\langle \hat{u}_1, \hat{u}_2 \rangle|, \quad \hat{\eta} \equiv |\langle \hat{v}_1, \hat{v}_2 \rangle| \asymp |\langle \hat{w}_1, \hat{w}_2 \rangle|.
\end{aligned}
\tag{5}
$$

The equivalences $|\langle \hat{u}_1, x_i \rangle| \asymp |\langle \hat{v}_1, y_i \rangle| \asymp |\langle \hat{w}_1, z_i \rangle|$, $|\langle \hat{v}_2, y_i \rangle| \asymp |\langle \hat{w}_2, z_i \rangle|$ and $|\langle \hat{v}_1, \hat{v}_2 \rangle| \asymp |\langle \hat{w}_1, \hat{w}_2 \rangle|$ are a direct consequence of our assumption (2), provided all mode dimensions of $\mathcal{T}_1$ are equal. We also highlight that $\hat{\theta}_{2i} \not\asymp \hat{\rho}_{2i}$ and $\hat{\kappa} \not\asymp \hat{\eta}$ in general, since the projection (3) is only applied to the first mode.

**Theoretical Alignments.** As $n \to \infty$, our main goal is to compute the asymptotic expectations of the singular values $\hat{\lambda}_i$ and alignments $\hat{\rho}_{1i}, \hat{\theta}_{2i}, \hat{\rho}_{2i}, \hat{\kappa}, \hat{\eta}$ at each deflation step $i$. Indeed, using concentration arguments, one can show that these quantities tend to concentrate around their respective expected values as $n$ grows large with variances of order $O(n^{-1})$, in the same vein as Benaych-Georges et al. (2011) which studied the fluctuations of the largest eigenvalues of large random matrices.

**Joint Estimation of Model Parameters.** We address the problem of jointly estimating the underlying model parameters, namely the signal-to-noise ratios $\beta_1, \beta_2$, and the correlation parameter $\alpha$ based on a **single realization** of $\mathcal{T}_1$. This allows us to construct an improved deflation algorithm in the correlated case. Our analysis partly relies on a recently developed random tensor theory approach (Seddik et al., 2021). In particular, we decompose the random tensor model obtained at each deflation step as in the following

---
**Generalized Deflation Step Theoretical Analysis Procedure** (Detailed Insight in Appendix G).

1. Identify the tensor model's **corresponding random matrix model**.
2. Describe the **limiting spectral measure** of the corresponding random matrix.
3. Compute the **asymptotic singular value and corresponding alignments**.
---

Next, we present a detailed analysis of step part of our approach.

## 3.2 FIRST DEFLATION STEP

Following the aforementioned macro steps, the first deflation step is performed in the following

### 3.2.1 CORRESPONDING RANDOM MATRIX MODEL & LIMITING SPECTRAL MEASURE

Starting from the best rank-1 approximation identities (4) for $i = 1$, it has been shown in Seddik et al. (2021) that the study of the random tensor $\mathcal{T}_1$ and its associated singular value and vectors $(\hat{\lambda}_1, \hat{u}_1, \hat{v}_1, \hat{w}_1)$ boils down to the analysis of the following block-wise contraction random matrix of size $n \times n$ (see Appendix D.1)

$$
\boldsymbol{N} \equiv \frac{1}{\sqrt{n}} \begin{pmatrix} 0 & \mathcal{W}(\hat{w}_1) & \mathcal{W}(\hat{v}_1) \\ \mathcal{W}(\hat{w}_1)^\top & 0 & \mathcal{W}(\hat{u}_1) \\ \mathcal{W}(\hat{v}_1)^\top & \mathcal{W}(\hat{u}_1)^\top & 0 \end{pmatrix}
\tag{6}
$$

To characterize the limits of $\lambda_1$ and the alignments $\hat{\rho}_{1i}$ for $i \in [2]$ when $n \to \infty$, the analysis boils down to the computation of the Stieltjes transform of the limiting spectral measure of the random matrix $\boldsymbol{N}$. We prove the latter statement in Appendix D.1.2. To compute the Stieltjes transform, the following technical assumption is needed. We provide more insight into that in the next section.

**Assumption 3.1** ("Recoverability"). As $n \to \infty$, there exists a sequence of critical points $(\hat{\lambda}_1, \hat{u}_1, \hat{v}_1, \hat{w}_1)$ such that $\hat{\lambda}_1 \xrightarrow{\text{a.s.}} \lambda_1 > 2\sqrt{\frac{2}{3}}$ and $\hat{\rho}_{1i} \xrightarrow{\text{a.s.}} \rho_{1i} > 0$.

Now, we finally characterize the limiting spectral measure of the random matrix $\boldsymbol{N}$ in the following theorem.

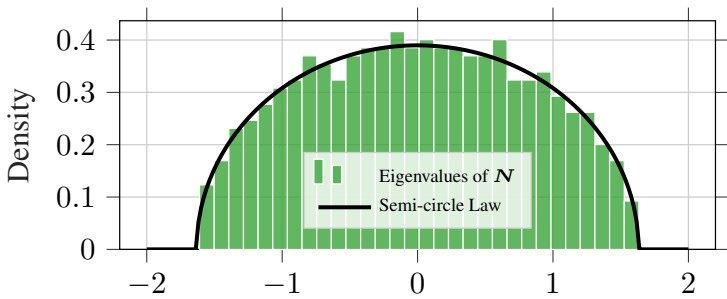

Figure 2: Spectrum of Random Matrix Model at **First Deflation Step N** (6) & Limiting Semi-Circle Law (Theorem 1)- One realization of $\mathcal{T}_1$ with Parameters: $p = 200$, $\beta_1 = 20$, $\beta_2 = 15$, $\alpha = 0.8$.

---

**Theorem 1.** *(Seddik et al., 2021, Corollary 1) Under Assumption 3.1, the spectral measure of* $N$ *converges weakly to a semi-circle law* $\mu$ *of compact support* $\left[-2\sqrt{\frac{2}{3}}, 2\sqrt{\frac{2}{3}}\right]$ *and density function* $\mu(dx) = \frac{3}{4\pi}\sqrt{\left(x^2 - \frac{8}{3}\right)^+}$. *Moreover, the Stieltjes transform of* $\mu$ *is*

$$r(z) = \frac{3}{4}\left(-z + \sqrt{z^2 - \frac{8}{3}}\right), \quad for \quad z > 2\sqrt{\frac{2}{3}}.$$

---

**Visualization of Theorem 1.** Figure 2 depicts the histogram of the eigenvalues of $N$ and the corresponding limiting semi-circle law as specified by Theorem 1. Note that the spectral measure of $N$ is not affected by the parameters $\beta_1, \beta_2$ and $\alpha$, but some conditions on them are required to ensure Assumption 3.1 ("Recoverability").

### 3.2.2 ASYMPTOTIC SINGULAR VALUE AND ALIGNMENTS

Following Theorem 1, by taking the expectation with respect to $\mathcal{W}$ of the identity $\hat{\lambda}_1 = \mathcal{T}_1(\hat{u}_1, \hat{v}_1, \hat{w}_1)$ (4), one can derive the following result.

**Proposition 3.2.** *The limiting singular value* $\lambda_1$ *satisfies the following*

$$\lambda_1 + r(\lambda_1) = \sum_{i=1}^{2} \beta_i \rho_{1i}^3. \tag{7}$$

*Proof.* See Appendix D.1.2. □

**Insight into Assumption 3.1.** Since the Stieltjes transform $r$ has to be evaluated at $\lambda_1$, the latter must lie outside the support of $\mu$, which is ensured by Assumption 3.1 if the signal strengths $\beta_1$ or $\beta_2$ are sufficiently high. In the absence of inter-component correlations, i.e $\alpha = 0$, it was shown in (Seddik et al., 2021, Corollary 3) that $\max\{\beta_1, \beta_2\}$ must be greater than $\frac{2\sqrt{3}}{3}$ to ensure $\lambda_1 > 2\sqrt{\frac{2}{3}}$. When $\lambda_1 \leq 2\sqrt{\frac{2}{3}}$, i.e., $\lambda_1$ lies inside the support of $\mu$, it basically corresponds to the case where the tensor $\mathcal{T}_1$ is indistinguishable from its noise counterpart $\mathcal{W}$, and hence recovering the signal components is information-theoretically impossible (Richard & Montanari, 2014; Lesieur et al., 2017; Jagannath et al., 2020; Goulart et al., 2021; Seddik et al., 2021).

Now, taking the expectation with respect to $\mathcal{W}$ of the remaining identities in (4, for $i = 1$) projected onto the signal components $x_i, y_i, z_i$ for $i \in [2]$, allows us to obtain Theorem 2. This result **exactly** characterizes the asymptotic behavior at the first deflation step.

---

**Theorem 2.** *Under Assumption 3.1, the limiting singular value* $\lambda_1$ *and corresponding alignments* $\rho_{1i}$ *for* $i \in [2]$ *of the first deflation step satisfy the following equations:*

$$f_r(\lambda_1) = \sum_{i=1}^{2} \beta_i \rho_{1i}^3, \quad h_r(\lambda_1)\rho_{1j} = \sum_{i=1}^{2} \beta_i \alpha_{ij} \rho_{1i}^2 \quad for \quad j \in [2], \tag{8}$$

---

> *where we recall $\alpha_{ij} = \alpha$ if $i \neq j$ and 1 otherwise, and we denoted $f_r(z) = z + r(z)$ and $h_r(z) = -\frac{1}{r(z)}$, with $r$ the Stieltjes transform of the corresponding random matrix model after deflation step 1 (semi-circle law, as depicted in Figure 3) as per Theorem 1.*

*Proof.* See D.1.2 and D.1.3. □

### 3.3 SECOND DEFLATION STEP

We henceforth turn to the description of the second deflation step asymptotics.

#### 3.3.1 CORRESPONDING RANDOM MATRIX MODEL & LIMITING SPECTRAL MEASURE

Denote $\hat{u}_3 = \hat{u}_2 - \gamma \langle \hat{u}_1, \hat{u}_2 \rangle \hat{u}_1$. We show in Appendix D.2 that the study of the second deflation step boils down to the analysis of the following $n \times n$ block-wise contraction random matrix

$$\boldsymbol{M} \equiv \frac{1}{\sqrt{n}} \begin{pmatrix} 0 & \mathcal{W}(\hat{w}_2) & \mathcal{W}(\hat{v}_2) \\ \mathcal{W}(\hat{w}_2)^\top & 0 & \mathcal{W}(\hat{u}_3) \\ \mathcal{W}(\hat{v}_2)^\top & \mathcal{W}(\hat{u}_3)^\top & 0 \end{pmatrix}, \tag{9}$$

Now, we demonstrate that for some $\gamma \neq 1$, the limiting spectral measure of $\boldsymbol{M}$ does **not** follow a semi-circle law due to the additional term $\gamma \langle \hat{u}_1, \hat{u}_2 \rangle \mathcal{W}(\hat{u}_1)$ induced by the correlation between the singular vectors $\hat{u}_1$ and $\hat{u}_2$. When $\gamma = 0$ or $\gamma = 1$, the term $\gamma \langle \hat{u}_1, \hat{u}_2 \rangle \mathcal{W}(\hat{u}_1)$ vanishes, in which cases the limiting spectral measure of $\boldsymbol{M}$ reduces to the semi-circle law in Theorem 1. For $\gamma = 1$, the identity $\mathcal{T}_2(\cdot, \hat{v}_2, \hat{w}_2) = \hat{\lambda}_2 \hat{u}_2$ in (4) with $\mathcal{T}_2$ (as in 3) implies

$$\lambda_2 \langle \hat{u}_1, \hat{u}_2 \rangle = \mathcal{T}_2(\hat{u}_1, \hat{v}_2, \hat{w}_2) = \mathcal{T}_1(\hat{u}_1, \hat{v}_2, \hat{w}_2) - \underbrace{\langle \hat{u}_1, \hat{u}_1 \rangle}_{=1} \mathcal{T}_1(\hat{u}_1, \hat{v}_2, \hat{w}_2) = 0 \tag{10}$$

which in turn directly implies $\langle \hat{u}_1, \hat{u}_2 \rangle = 0$ since the spectral norm of the tensor $\mathcal{T}_2$ is non-zero, due to the presence of the noise term. We therefore provide the result characterizing the limiting spectral measure of $\boldsymbol{M}$ for any $\gamma \in [0, 1]$, and which in turn generalizes Theorem 1 to random contraction matrices of the form in equation 9. We start by the following definition.

**Definition 3.3.** Let $\nu$ be the probability measure with Stieltjes transform $q(z) = a(z) + 2b(z)$ verifying $\Im[q(z)] > 0$ for $\Im[z] > 0$, where $a(z)$ and $b(z)$ satisfy the following equations, for $z \notin \mathrm{supp}(\nu)$

$$[2b(z) + z]\, a(z) + \frac{1}{3} = 0, \quad (a(z) + z - \tau b(z))b(z) + \frac{1}{3} = 0, \tag{11}$$

for some parameter $\tau \in \mathbb{R}$. Moreover, the density function corresponding to $\nu$ is given by the Stieltjes inverse formula $\nu(dx) = \frac{1}{\pi} \lim_{\varepsilon \to 0} \Im[q(x + i\varepsilon)]$.

Akin to the analysis of the first deflation step, we need additional technical assumptions to describe the limiting singular value $\lambda_2$ and corresponding alignments.

**Assumption 3.4.** As $n \to \infty$, there exists a sequence of critical points $(\hat{\lambda}_2, \hat{u}_2, \hat{v}_2, \hat{w}_2)$ such that, for $i \in [2]$

$$\hat{\lambda}_2 \xrightarrow{\text{a.s.}} \lambda_2, \; \hat{\theta}_{2i} \xrightarrow{\text{a.s.}} \theta_{2i}, \; \hat{\kappa} \xrightarrow{\text{a.s.}} \kappa, \; \hat{\rho}_{2i} \xrightarrow{\text{a.s.}} \rho_{2i}, \; \hat{\eta} \xrightarrow{\text{a.s.}} \eta,$$

where $\lambda_2 \notin \mathrm{supp}(\nu)$ with $\nu$ defined in Definition 3.3 for $\tau = \gamma \kappa^2 - 1 + \kappa(\gamma - 1)$ and suppose $\theta_{2i}, \kappa, \rho_{2i}, \eta > 0$.

We therefore have the following theorem which characterizes the limiting spectral measure of $\boldsymbol{M}$.

> **Theorem 3.** *Under Assumption 3.4, the spectral measure of $\boldsymbol{M}$ converges weakly to the probability measure $\nu$ defined in Definition 3.3 for $\tau = \gamma \kappa^2 - 1 + \kappa(\gamma - 1)$.*

*Proof.* See D.2.1. □

**Interpretation & Visualization of Theorem 3.** In essence, if the involved alignments in the second deflation step converge asymptotically, Theorem 3 states the convergence of the spectral measure of $\boldsymbol{M}$ to the deterministic measure $\nu$ defined in Definition 3.3 for $\tau = \gamma \kappa^2 - 1 + \kappa(\gamma - 1)$. We particularly recall that $\kappa$ corresponds to the limit of $\langle \hat{u}_1, \hat{u}_2 \rangle$, which highlights the fact that the spectrum of $\boldsymbol{M}$ can be deformed if the singular vectors $u_1$ and $u_2$ are correlated, i.e., if $\gamma \neq 1$. This phenomenon is depicted in Figure 3 where we see that for $\gamma = 0.85$, the limiting spectral measure of

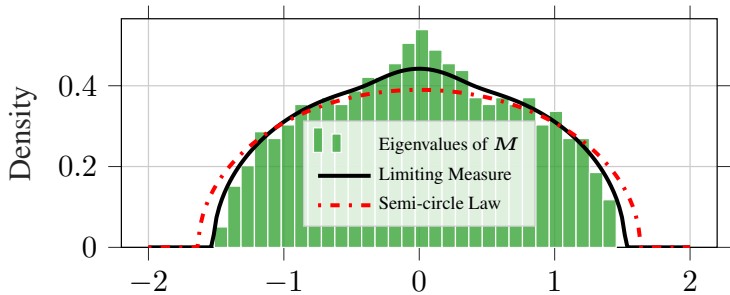

Figure 3: Spectrum of Random Matrix Model at **Second Deflation Step** M (9) & Limiting (Theoretically Obtained) Law (3)- One realization of $\mathcal{T}_1$ with Parameters: $p = 200$, $\beta_1 = 20$, $\beta_2 = 15$, $\alpha = 0.8$, $\gamma = 0.85$.

$$
\begin{cases}
f_q(\lambda_2) - \frac{\gamma\kappa\eta^2}{3}r(\lambda_1) - 2\gamma\kappa^2 b(\lambda_2) = \sum_{i=1}^2 \beta_i\theta_{2i}\rho_{2i}^2 - \gamma\kappa\sum_{i=1}^2 \beta_i\rho_{1i}\rho_{2i}^2, \\
[f_q(\lambda_2) - a(\lambda_2)]\theta_{2j} - \gamma\rho_{1j}\left[\frac{\eta^2}{3}r(\lambda_1) + 2\kappa b(\lambda_2)\right] = \sum_{i=1}^2 \beta_i\alpha_{ij}\rho_{2i}^2 - \gamma\rho_{1j}\sum_{i=1}^2 \beta_i\rho_{1i}\rho_{2i}^2 \quad \text{for} \quad j \in [2], \\
[\lambda_2 + 2(1-\gamma)b(\lambda_2)]\kappa = (1-\gamma)\left[\sum_{i=1}^2 \beta_i\rho_{1i}\rho_{2i}^2 - \frac{\eta^2}{3}r(\lambda_1)\right], \\
[f_q(\lambda_2) - (1+\gamma\kappa^2)b(\lambda_2)]\rho_{2j} = \sum_{i=1}^2 \beta_i\theta_{2i}\rho_{2i}\alpha_{ij} - \gamma\kappa\left[\sum_{i=1}^2 \beta_i\rho_{1i}\rho_{2i}\alpha_{ij} - \frac{\rho_{1j}\eta}{3}r(\lambda_1)\right] \quad \text{for} \quad j \in [2], \\
[\lambda_2 + a(\lambda_2) + (1-\gamma\kappa^2)b(\lambda_2) - \frac{\gamma\kappa}{3}r(\lambda_1)]\eta = \sum_{i=1}^2 \beta_i\theta_{2i}\rho_{1i}\rho_{2i} - \gamma\kappa\sum_{i=1}^2 \beta_i\rho_{1i}^2\rho_{2i}.
\end{cases}
\tag{13}
$$

$M$ is no longer described by the semi-circle law. On the contrary, if $\gamma = 1$ we have $\kappa = 0$ as we saw in 9, which implies that $\tau = -1$. In this case, the limiting spectral measure $\nu$ becomes equal to $\mu$, thereby describing again a semi-circle law. This can be trivially checked from Definition 3.3 by setting $\tau = -1$ and $a(z) = b(z)$, and we therefore find $a(z) = b(z) = \frac{r(z)}{3}$ and $q(z) = r(z)$. Note that for $\gamma \in (0, 1)$, the Stieltjes transform $q(z)$ can be computed numerically by alternating the equations in equation 11 as per Algorithm 1, which can be proved to converge to a fixed point in the same vein as in Louart & Couillet (2018).

### 3.3.2 ASYMPTOTIC SINGULAR VALUE AND ALIGNMENTS

As for the first deflation step, taking the expectation w.r.t. $\mathcal{W}$ of the identity $\hat{\lambda}_2 = \mathcal{T}_2(\hat{u}_2, \hat{v}_2, \hat{w}_2)$ in equation 4 allows us to obtain the equation satisfied by $\lambda_2$, see Appendix D.2.2, which yields

$$
f_q(\lambda_2) - \frac{\gamma\kappa\eta^2}{3}r(\lambda_1) - 2\gamma\kappa^2 b(\lambda_2) = \sum_{i=1}^2 \beta_i\theta_{2i}\rho_{2i}^2 - \gamma\kappa\sum_{i=1}^2 \beta_i\rho_{1i}\rho_{2i}^2,
\tag{12}
$$

where $f_q(z) = z + q(z)$. Again, the limiting singular value $\lambda_2$ must lie outside the support of $\nu$, as we assumed in Assumption 3.4, since its corresponding Stieltjes transform $q(z)$ (and the function $b(\cdot)$) needs to be evaluated at $\lambda_2$. In fact, if $\lambda_2 \in \text{supp}(\nu)$, then it is information-theoretically impossible to recover the second signal term (i.e., the one with strength $\min\{\beta_1, \beta_2\}$).

Moreover, taking the expectation w.r.t. $\mathcal{W}$ of the remaining identities in equation 4 for $i = 2$, projected on the signal components $x_i, y_i, z_i$ for $i \in [2]$ and the first singular vectors $u_1, v_1, w_1$, allows us to derive the result characterizing the behavior of the second deflation step.

> **Theorem 4.** *Under Assumption 3.4, the limiting singular value $\lambda_2$ and corresponding alignments $\theta_{2i}, \rho_{2i}$ for $i \in [2]$ and $\kappa, \eta$ of the second deflation step satisfy the system of equations in equation 13.*

*Proof.* See D.2.2 and D.2.3. □

### 3.4 MAIN ALGORITHM SKETCH

We are now in place to describe the proposed $\gamma$-orthogonalized tensor deflation algorithm. Our principal insight lies in the fact that, for $\beta_1 > \beta_2$ for instance, the asymptotic alignments $\theta_{22}$ and $\rho_{22}$ at the second deflation step are **concave** functions of the parameter $\gamma$ (as depicted in Figure 9 for $\alpha = 0.6$ and Figure 10 for different values of $\alpha$ in Appendix E.4). Therefore, there exists an optimal value $\gamma^*$ which maximizes such alignments and which we need to **jointly tune** in order to recover

the signal components efficiently, given a **single** realization of the spiked random tensor $\mathcal{T}_1$. To this end, we motivate our algorithmic choices in the following procedure

- First we perform a standard orthogonalized tensor deflation with $\gamma = 1$ which corresponds to the steps 1 and 2 of Algorithm 2.

- Then we estimate the underlying model parameters, i.e., $\beta_1, \beta_2$, and $\alpha$ as we discussed in Appendix E.2. This corresponds to steps 3 and 4 of Algorithm 2.

- In order to find the optimal parameter $\gamma^*$ which maximizes the alignment $\hat{\rho}_{22}$ for instance. We update $\gamma$ as $\gamma \leftarrow \gamma - \epsilon$ for some small step size $\epsilon > 0$ and starting from $\gamma = 1$, while solving the system of equations (13) to get an estimate of $\hat{\rho}_{22}$. We stop updating $\gamma$ when the maximum value of $\hat{\rho}_{22}$ is reached and we return the corresponding maximizer $\gamma^*$. Note that at each iteration, the system (equation 13) is solved numerically and initialized with the previous iteration estimates. This corresponds to steps 5-12 in Algorithm 2.

- Then, we perform orthogonalized deflation with $\gamma^*$ along the modes 1 and 2, which provides a better estimation of the signal component denoted as $\hat{\lambda}_2 \hat{u}_2^* \otimes \hat{v}_2^* \otimes \hat{w}_2^*$. This corresponds to steps 13 and 14 of Algorithm 2.

- Finally, in step 15 of Algorithm 2, we re-estimate the first signal component by performing a best rank-1 approximation of $\mathcal{T}_1 - \min\{\hat{\beta}_1, \hat{\beta}_2\} \hat{u}_2^* \otimes \hat{v}_2^* \otimes \hat{w}_2^*$ with $\hat{\beta}_1, \hat{\beta}_2$ being the estimated SNRs from step 4 of Algorithm 2.

Given the space constraints, we present the full $\gamma$-orthogonalized tensor deflation algorithm pseudo-code in Algorithm 2.

## 4    EXPERIMENTAL SETUP & EVALUATION

In the following, we list a few representative experiments. In addition, we provide a detailed set of supplementary experiments in Appendix E.

**Experimental Setup.** We operate in the synthetic setting, where we randomly generate a low-rank spiked tensor model (as in 2.1) starting from a single tensor component sampled from a Gaussian distribution (we provide more details and a code sample in E.1). We evaluate our proposed algorithm's recovery quality on the algorithmic performance (final deflation alignments & single-realization estimation quality of the model parameters), on estimation robustness compared to state-of-the-art (averaging over multiple instances, variance statistics) and how well our developed theory corresponds to the performance in practice.

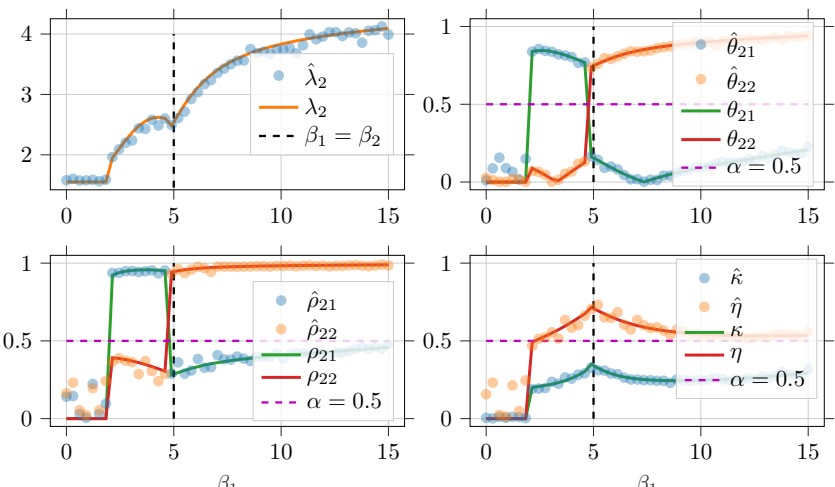

Figure 4: **Simulated** vs **theoretical** asymptotic singular value and alignments corresponding to the second deflation step as per Theorem 4. We considered $\beta_2 = 5$, $\alpha = 0.5$, $p = 100$, $\gamma = 0.8$ and varying $\beta_1 \in [0, 15]$. The system of equations (13) is solved numerically and initialized with the simulated singular value and alignments (dotted curves) from a single realization of $\mathcal{T}_1$.

**Simulated vs Theoretical Singular Value and Alignments.** Figure 4 depicts the simulated singular value and alignments between the different signal components at the second deflation step along with their theoretical asymptotic counterparts as given by Theorem 4. We observe that our developed theory accurately describes the expected behavior and thereby offers strong theoretical guarantees on the performance of our proposed algorithm. On an important note, we further stress out that for a fixed (large enough) $\beta_2$ and $\alpha \neq 1$, there exists a threshold for $\beta_1$ below which it is information-theoretically impossible to recover the second signal component. This behavior is already visible from Figure 4 for $\beta_1 \approx 2$, below which all the alignments are asymptotically zero and the asymptotic singular value converges to the right edge of the distribution $\nu$. This exactly corresponds to the scenario where Assumption 3.4 is not verified.

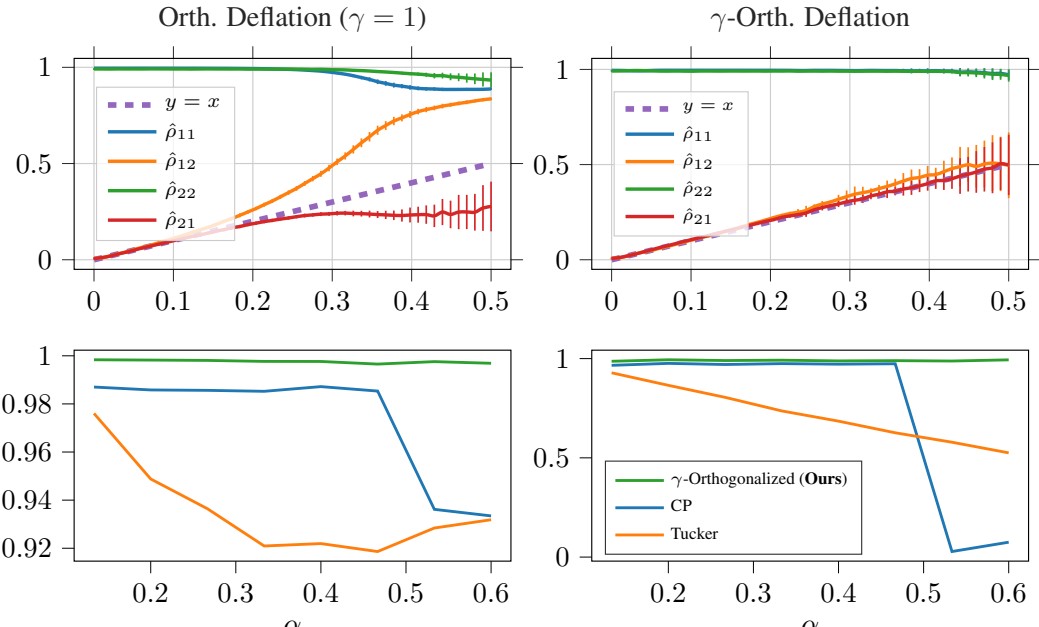

Figure 5: **(Top): Alignments of first and second deflation steps** in function of $\alpha$. **(Top Left)** Performance of standard orthogonalized deflation ($\gamma = 1$) and **(Top Right)** of the $\gamma$-orthogonalized deflation procedure, as per Algorithm 2. (Averaged over) 200 realizations of $\mathcal{T}_1$: $\beta_1 = 6$, $\beta_2 = 5.7$ and $p = 150$. **(Bottom) Alignment at convergence** in function of inter-component correlation $\alpha$. **(Bottom Left)** Alignments on mode 1 projections, **(Bottom Right)** Alignments on modes 2 and 3 (equal by virtue of 5). (Averaged over) 200 realizations of $\mathcal{T}_1$: $\beta_1 = 10$, $\beta_2 = 8$ and $p = 150$.

**Benchmarks.** Figure 5 showcases a two-level benchmarking. Figures on **(Top)** compare the performances of the standard orthogonalized deflation ($\gamma = 1$) **(Top Left)** and our proposed $\gamma$-orthogonalized version **(Top Right)**, while varying the signal correlation parameter $\alpha$. This allows us to measure the effect of correlation in orthogonalized deflation procedures and corresponds exactly to our considered setting and applications. As theoretically anticipated, we observe a clear advantage of our proposed approach with estimated alignments $\hat{\rho}_{11}$ and $\hat{\rho}_{22}$ very close to 1, thus accurately estimating the different signal components at different levels of correlation $\alpha$. For its vanilla orthogonalized deflation counterpart, in addition to decaying alignments $\hat{\rho}_{11}$ and $\hat{\rho}_{22}$ (e.g., $\alpha \geq 0.3$), we also observe a value close to 1 of $\hat{\rho}_{12}$ thus showing a high level of correlation between the estimation of different signal components at the first deflation step. On the **(Bottom)** part, we compare the alignments for mode 1 **(Bottom Left)** and modes 2, 3 (equal by virtue of 5) **(Bottom Right)** of our proposed algorithm to prevalent state-of-the-art spectral tensor decomposition algorithms, namely CANDECOMP/PARAFAC (CP) and Tucker. We see a very clear advantage of the $\gamma$-orthogonalized approach, as the other approaches collapse for higher correlation levels.

## 5 PERSPECTIVE

We address some of the limitations of our work and elaborate on potential promising directions, including **Extension to Higher Ranks & Orders, Evaluation on Real-World Data** and **Other Theoretical Considerations**. We treat these directions in depth in Appendix G.

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

# Appendix

## Table of Contents

# A    RELATED WORK

As a first step towards understanding the behavior of tensor methods, Richard & Montanari (2014) introduced the concept of tensor PCA by studying the so-called *spiked tensor model* of the form $\beta x^{\otimes d} + \mathcal{W}/\sqrt{p}$ where $x \in \mathbb{R}^p$ is a high-dimensional vector of the unit norm which represents the (rank-1) signal of interest, $\mathcal{W}$ is a symmetric Gaussian random noise tensor of order $d$, and $\beta \geq 0$ is a parameter controlling the signal strength. This statistical model raises many fundamental questions which mainly concern the theoretical and algorithmic guarantees that ensure the efficient recovery of the hidden signal $x$.

A volume of works focused on addressing these questions (Perry et al., 2020; Lesieur et al., 2017; Jagannath et al., 2020; Chen et al., 2021; Goulart et al., 2021; Auddy & Yuan, 2022; Ben Arous et al., 2021). The first main result was for tensors of order $d \geq 3$, for which it was shown that there exists a statistical threshold $\beta_{stat}$ of $O(1)$ in the tensor dimension, which defines the information-theoretic limit above which signal recovery is possible, with the maximum likelihood estimator (MLE), and below which signal recovery is impossible.

While recovery above $\beta_{stat}$ is theoretically possible from an information-theoretic standpoint, solving the underlying MLE problem remains NP-hard in the worst case (Hillar & Lim, 2013). Indeed, Richard & Montanari (2014) suggest through heuristics that there exists an algorithmic threshold $\beta_{algo} = O(p^{\frac{d-2}{4}})$ beyond which recovery is possible with a polynomial time algorithm, therefore implying the existence of a theoretical-algorithmic spectral gap where no polynomial time algorithm has been proven to be efficient in signal recovery. These suggestions were rigorously proven (Lesieur et al., 2017; Jagannath et al., 2020; Chen et al., 2021; Huang et al., 2022) and generalized to non-symmetric tensors (Ben Arous et al., 2021; Seddik et al., 2021; Auddy & Yuan, 2022).

From a practical standpoint, to be able to unleash the full potential of tensor methods, higher low-rank (beyond rank-1) signal reconstruction problems need to be considered, thus motivating the study of *low-rank spiked tensor models*. In particular, and in a more realistic scenario, one would be interested in extracting low-rank hidden structure from random noise, for which the model can naturally be extended to $\sum_{i=1}^{r} \beta_i x_i^{\otimes d} + \mathcal{W}/\sqrt{p}$ where $r$ is the rank of the signal of interest.

In this line of work, Chen et al. (2021) prove that the asymptotic behavior of a low-rank spiked tensor model with orthogonal signal components, i.e., $\langle x_i, x_j \rangle = 0$ for $i \neq j \in [r]$, can be understood from the analysis of a rank-1 model. Moreover, da Silva et al. (2015b;a) show that estimating a higher rank signal boils down to performing successive rank-1 approximations using iterative *tensor deflation*. While the latter result provides a more tractable approach to low-rank signal recovery in the orthogonal case, it proves to be inefficient (in general) in signal reconstruction in the non-orthogonal case (Seddik et al., 2022).

Other enhanced deflation techniques rely on orthogonal projections (Mackey, 2008) while exhibiting the same behavior as the standard deflation in the non-orthogonal case, as illustrated in Figure 1. The latter depicts signal recovery, as measured by alignments of the recovered signal with the hidden low-rank signal, from a simple noiseless rank-two tensor $\sum_{i=1}^{2} \beta_i x_i^{\otimes 3}$ using orthogonalized deflation (Mackey, 2008). We clearly observe that when $\beta_1 \approx \beta_2$, the non-orthogonality of $x_1$ and $x_2$ prevents efficient recovery. We highlight the fact that measuring alignments is a concrete performance measure of recovery quality, as mentioned in 2.1.

# B    A REVIEW OF RANDOM MATRIX/TENSOR THEORY

We aim to provide a self-contained review of some key results of random matrix/tensor theory (RMT/RTT) in the following.

## B.1    TENSOR CONCEPTS

**3-order tensors.**    The set of 3-order tensors of mode dimension $d$ is denoted $\mathbb{R}^{d \times d \times d}$. The scalar $T_{ijk}$ or $[\mathcal{T}]_{ijk}$ denotes the $(i, j, k)$'th entry of a tensor $\mathcal{T} \in \mathbb{R}^{d \times d \times d}$. In the remainder, we will mainly consider tensors from $\mathbb{R}^{d \times d \times d}$, and for brevity, we may omit the notation $\mathcal{T} \in \mathbb{R}^{d \times d \times d}$ unless specified otherwise.

**Rank-$r$ tensors.**    A tensor $\mathcal{T}$ is said to be of rank-1 if it can be represented as the outer product of three real-valued vectors $x, y, z \in \mathbb{R}^d$. In this case, we write $\mathcal{T} = x \otimes y \otimes z$, where the outer product is defined as $[x \otimes y \otimes z]_{ijk} = x_i y_j z_k$. More generally, a tensor $\mathcal{T}$ is said to be of rank-$r$, for some integer $r$, if it can be expressed as the sum of $r$ rank-1 terms, written as $\mathcal{T} = \sum_{i=1}^{r} x_i \otimes y_i \otimes z_i$,

where $x_i, y_i, z_i \in \mathbb{R}^d$ for all $i \in [r]$. To maintain consistency, we will adhere to the convention of using $x_i$ or $u_i$ to represent the components of the first mode, $y_i$ or $v_i$ to represent the components of the second mode, and $z_i$ or $w_i$ to represent the components of the third mode throughout the paper.

**Tensor contractions.** The first mode contraction of a tensor $\mathcal{T}$ with a vector $x$ results in a matrix denoted $\mathcal{T}(x, \cdot, \cdot)$ with entries $[\mathcal{T}(x, \cdot, \cdot)]_{jk} = \sum_{i=1}^d x_i T_{ijk}$. Similarly, $\mathcal{T}(\cdot, y, \cdot)$ and $\mathcal{T}(\cdot, \cdot, z)$ denote the second and third mode contractions of $\mathcal{T}$ with vectors $y$ and $z$ respectively. We will sometimes denote these contractions by $\mathcal{T}(x)$, $\mathcal{T}(y)$, and $\mathcal{T}(z)$ if there is no ambiguity. The contraction of $\mathcal{T}$ on two vectors $x, y$ is a vector denoted $\mathcal{T}(x, y, \cdot)$ with entries $[\mathcal{T}(x, y, \cdot)]_k = \sum_{ij} x_i y_j T_{ijk}$. The contraction of $\mathcal{T}$ on three vectors $x, y, z$ is a scalar denoted $\mathcal{T}(x, y, z) = \sum_{ijk} x_i y_j z_k T_{ijk}$. The first mode contraction of $\mathcal{T}$ with a matrix $\boldsymbol{M} \in \mathbb{R}^{d \times d}$ results in a tensor denoted $\mathcal{T} \times_1 \boldsymbol{M}$ with entries $[\mathcal{T} \times_1 \boldsymbol{M}]_{ijk} = \sum_{i'=1}^d M_{ii'} T_{i'jk}$. Similarly, $\mathcal{T} \times_2 \boldsymbol{N}$ and $\mathcal{T} \times_3 \boldsymbol{P}$ denote the second and third modes tensor-matrix contractions of the tensor $\mathcal{T}$ with the matrices $\boldsymbol{N}$ and $\boldsymbol{P}$ respectively. The notation $u \otimes \boldsymbol{M}$ stands for the tensor with entries $u_i M_{jk}$.

**Tensor norms.** The Frobenius norm of a tensor $\mathcal{T}$ is denoted $\|\mathcal{T}\|_F$ with $\|\mathcal{T}\|_F^2 = \sum_{ijk} T_{ijk}^2$. The spectral norm of $\mathcal{T}$ is $\|\mathcal{T}\| = \sup_{u,v,w \in \mathbb{S}^{d-1}} |\mathcal{T}(u, v, w)|$.

**Spectral Normalization.** We highlight that the normalization of the noise tensor, i.e $\mathcal{W}$ in equation 1, by the sum of mode dimension, i.e $n$ in equation 1, ensures that the spectral norm of $\mathcal{T}$ is of $O(1)$ with respect to mode dimensions $p_i$. This follows from a standard concentration result (Seddik et al. (2021), Lemma 4).

**Best rank-1 approximation and tensor power iteration.** A best rank-1 approximation of $\mathcal{T}$ corresponds to a rank-1 tensor $\lambda u \otimes v \otimes w$, where $\lambda > 0$ and $u, v, w$ are unitary vectors, that minimizes the square loss $\|\mathcal{T} - \lambda u \otimes v \otimes w\|_F^2$. The latter generalizes to tensors the concept of singular value and vectors (Lim, 2005) and the scalar $\lambda$ coincides with the spectral norm of $\mathcal{T}$. In particular, the quadruple $(\lambda, u, v, w)$ satisfies the following identities

$$\begin{cases} \mathcal{T}(\cdot, v, w) = \lambda u, & \mathcal{T}(u, \cdot, w) = \lambda v, \\ \mathcal{T}(u, v, \cdot) = \lambda w, & \lambda = \mathcal{T}(u, v, w). \end{cases} \tag{14}$$

Such a best rank-1 approximation can be computed via *tensor power iteration* which consists in iterating

$$u \leftarrow \frac{\mathcal{T}(\cdot, v, w)}{\|\mathcal{T}(\cdot, v, w)\|} \quad v \leftarrow \frac{\mathcal{T}(u, \cdot, w)}{\|\mathcal{T}(u, \cdot, w)\|} \quad w \leftarrow \frac{\mathcal{T}(u, v, \cdot)}{\|\mathcal{T}(u, v, \cdot)\|}$$

starting from some initialization (Anandkumar et al., 2014).

## B.2 RANDOM MATRIX THEORY

**Resolvent.** Specifically, we will consider the *resolvent* formalism Hachem et al. (2007), which allows to characterize the spectral behavior of large symmetric random matrices. Given a symmetric matrix $\boldsymbol{S} \in \mathbb{R}^{n \times n}$, the resolvent of $\boldsymbol{S}$ is defined as $\boldsymbol{R}(z) = (\boldsymbol{S} - z\boldsymbol{I}_n)^{-1}$ for $z \in \mathbb{C} \setminus \mathrm{Sp}(\boldsymbol{S})$, where $\mathrm{Sp}(\boldsymbol{S})$ denotes the spectrum of matrix $\boldsymbol{S}$.

**Motivation.** In essence, RMT focuses on describing the distribution of eigenvalues of large random matrices. Typically, under certain technical assumptions on some random matrix $\boldsymbol{S} \in \mathbb{R}^{n \times n}$ with eigenvalues $\lambda_1, \ldots, \lambda_n$, the *empirical spectral measure* of $\boldsymbol{S}$, defined as $\hat{\mu} = \frac{1}{n} \sum_{i=1}^n \delta_{\lambda_i}$, converges in the weak sense Van Der Vaart & Wellner (1996) to some deterministic probability measure $\mu$ as $n \to \infty$ and RMT aims at describing such $\mu$. To this end, one of the widely considered approaches relies on the *Stieltjes transform* Tao (2012).

**Stieltjes Transform.** Given a probability measure $\mu$, the Stieltjes transform of $\mu$ is defined as $g_\mu(z) = \int \frac{d\mu(\lambda)}{\lambda - z}$ with $z \in \mathbb{C} \setminus \mathrm{Supp}(\mu)$, and the inverse formula allows one to describe the density function of $\mu$ as $\mu(dx) = \frac{1}{\pi} \lim_{\varepsilon \to 0} \Im[g_\mu(x + i\varepsilon)]$.

**Relating the Stieltjes Transform & the Resolvent.** The Stieltjes transform of the empirical spectral measure, $\hat{\mu}$, is closely related to the resolvent of $\boldsymbol{S}$ through the normalized trace operator. In fact, $g_{\hat{\mu}}(z) = \frac{1}{n} \mathrm{tr}\, \boldsymbol{R}(z)$ and the *almost sure* convergence of $g_{\hat{\mu}}(z)$ to some deterministic Stieltjes

transform $g(z)$ is equivalent to the weak convergence between the underlying probability measures Tao (2012). Our analysis relies on estimating quantities involving $\frac{1}{n} \operatorname{tr} \boldsymbol{R}(z)$, making the use of the resolvent approach a natural choice.

## C    KEY LEMMAS

In this section, we recall some key lemmas that are at the heart of our analysis.

**Lemma C.1** (Woodbury Matrix Identity). *Let $\boldsymbol{A} \in \mathbb{R}^{n \times n}$, $\boldsymbol{B} \in \mathbb{R}^{r \times r}$, $\boldsymbol{U} \in \mathbb{R}^{n \times r}$ and $\boldsymbol{V} \in \mathbb{R}^{r \times n}$, we have*

$$\left(\boldsymbol{A} + \boldsymbol{U}\boldsymbol{B}\boldsymbol{V}\right)^{-1} = \boldsymbol{A}^{-1} - \boldsymbol{A}^{-1}\boldsymbol{U}\left(\boldsymbol{B}^{-1} + \boldsymbol{V}\boldsymbol{A}^{-1}\boldsymbol{U}\right)^{-1}\boldsymbol{V}\boldsymbol{A}^{-1}$$

The following perturbation lemma is wildly used in RMT. Basically, it states that the normalized trace operator is invariant through low-rank perturbations in high dimension. The notation $a = O_n(b_n)$ means that $a$ is of order $O(b_n)$ as $n \to \infty$.

**Lemma C.2** (Perturbation Lemma, (Silverstein & Bai, 1995)). *Let $\boldsymbol{M} \in \mathbb{R}^{n \times n}$ and $\boldsymbol{P} \in \mathbb{R}^{n \times n}$ such that $\|\boldsymbol{M}\| = O_n(1)$, $\|\boldsymbol{P}\| = O_n(1)$ and $\operatorname{rank}(\boldsymbol{P}) = O_n(1)$. For all $z \in \mathbb{C} \setminus \operatorname{Sp}(\boldsymbol{M} + \boldsymbol{P})$, we have*

$$\frac{1}{n} \operatorname{tr}\left(\boldsymbol{M} + \boldsymbol{P} - z\boldsymbol{I}_n\right)^{-1} = \frac{1}{n} \operatorname{tr}\left(\boldsymbol{M} - z\boldsymbol{I}_n\right)^{-1} + O_n(n^{-1})$$

*Proof.* Simple consequence of the Woodbury identity from Lemma C.1 applied to the matrix $\boldsymbol{M} + \boldsymbol{P}$. $\qquad\square$

Our analysis will particularly rely on computing expectations which we drive through the classical Stein's Lemma.

**Lemma C.3** (Stein's Lemma, Stein (1981)). *Let $W \sim \mathcal{N}(0, \sigma^2)$ and $f$ some continuously differentiable function having at most polynomial growth, then*

$$\mathbb{E}\left[W f(W)\right] = \sigma^2 \mathbb{E}\left[f'(W)\right]$$

*when the above expectations exist.*

## D    PROOFS OF THE MAIN RESULTS

We recall our considered spiked tensor model as follows

$$\mathcal{T}_1 = \mathcal{S} + \frac{1}{\sqrt{n}}\mathcal{W} \in \mathbb{R}^{p \times p \times p} \quad \text{with} \quad \mathcal{S} = \sum_{i=1}^{2} \beta_i x_i \otimes y_i \otimes z_i \tag{15}$$

where $\|x_i\| = \|y_i\| = \|z_i\| = 1$, $\beta_i \geq 0$, $n = 3p$ and $W_{ijk} \sim \mathcal{N}(0, 1)$. In the remainder, if some quantity expresses as $a(n) = \sum_{i=1}^{r} b_i(n)$, the notation $a(n) \simeq b_j(n)$ means that $b_j(n)$ is the only contributing term in the expression of $a(n)$ as $n$ goes to infinity.

### D.1    FIRST DEFLATION STEP

The singular vectors $u_1, v_1$ and $w_1$ of $\mathcal{T}_1$ corresponding to its largest singular value $\lambda_1$ satisfy

$$\mathcal{T}_1(\cdot, v_1, w_1) = \lambda_1 u_1, \quad \mathcal{T}_1(u_1, \cdot, w_1) = \lambda_1 v_1, \quad \mathcal{T}_1(u_1, v_1, \cdot) = \lambda_1 w_1 \tag{16}$$

In the remainder, we will need to compute the derivatives of the singular vectors $u_1, v_1$ and $w_1$ w.r.t.the entries of the noise tensor $\mathcal{W}$. From (Seddik et al., 2021, Appendix B.1), we have

$$\begin{pmatrix} \frac{\partial u_1}{\partial W_{ijk}} \\ \frac{\partial v_1}{\partial W_{ijk}} \\ \frac{\partial w_1}{\partial W_{ijk}} \end{pmatrix} = -\frac{1}{\sqrt{n}}\left(\begin{pmatrix} 0 & \mathcal{T}_1(w_1) & \mathcal{T}_1(v_1) \\ \mathcal{T}_1(w_1)^\top & 0 & \mathcal{T}_1(u_1) \\ \mathcal{T}_1(v_1)^\top & \mathcal{T}_1(u_1)^\top & 0 \end{pmatrix} - \lambda_1\boldsymbol{I}_n\right)^{-1} \begin{pmatrix} v_{1j}w_{1k}(e_i - u_{1i}u_1) \\ u_{1i}w_{1k}(e_j - v_{1j}v_1) \\ u_{1i}v_{1j}(e_k - w_{1k}w_1) \end{pmatrix} \tag{17}$$

which results from deriving the identities in equation 16 w.r.t.the entry $W_{ijk}$ of the noise tensor $\mathcal{W}$. In particular, as demonstrated in Seddik et al. (2021), the only contributing terms in the quantities we will compute later on will depend only on traces of the resolvent matrix appearing in equation 17.

### D.1.1 Limiting Stieltjes transform

Since the tensor $\mathcal{T}_1$ is a low-rank perturbation of a random tensor $\mathcal{W}$, by Lemma C.2, the normalized trace of the resolvent in equation 17 is asymptotically equal to the normalized trace of the resolvent of the following random matrix

$$\boldsymbol{N} = \frac{1}{\sqrt{n}} \begin{pmatrix} 0 & \mathcal{W}(w_1) & \mathcal{W}(v_1) \\ \mathcal{W}(w_1)^\top & 0 & \mathcal{W}(u_1) \\ \mathcal{W}(v_1)^\top & \mathcal{W}(u_1)^\top & 0 \end{pmatrix} \tag{18}$$

Let $R(z) = (\boldsymbol{N} - z\boldsymbol{I}_n)^{-1}$ be the corresponding resolvent. We denote the different sub-blocks of $R(z)$ as

$$R(z) = \begin{pmatrix} R^{11}(z) & R^{12}(z) & R^{13}(z) \\ R^{12}(z)^\top & R^{22}(z) & R^{23}(z) \\ R^{13}(z)^\top & R^{23}(z)^\top & R^{33}(z) \end{pmatrix} \tag{19}$$

It has been shown in (Seddik et al., 2021, Appendix B.2) that

$$\frac{1}{n} \operatorname{tr} R^{ii}(z) \xrightarrow[n\to\infty]{} r_i(z) = \frac{r(z)}{3} \quad \text{and} \quad \frac{1}{n} \operatorname{tr} R(z) \xrightarrow[n\to\infty]{} r(z) \tag{20}$$

with

$$\boxed{r(z) = \frac{3}{4}\left(-z + \sqrt{z^2 - \frac{8}{3}}\right)} \tag{21}$$

### D.1.2 Estimation of the singular value

**Estimation of $\lambda_1$:** From the identities in equation 16, we have

$$\lambda_1 = \mathcal{T}_1(u_1, v_1, w_1) = \mathcal{S}(u_1, v_1, w_1) + \frac{1}{\sqrt{n}}\mathcal{W}(u_1, v_1, w_1)$$

and

$$\frac{1}{\sqrt{n}}\mathbb{E}\left[\mathcal{W}(u_1, v_1, w_1)\right] = \frac{1}{\sqrt{n}} \sum_{ijk} \mathbb{E}[u_{1i}v_{1j}w_{1k}W_{ijk}]$$

$$= \frac{1}{\sqrt{n}} \sum_{ijk} \mathbb{E}\left[\frac{\partial u_{1i}}{\partial W_{ijk}}v_{1j}w_{1k}\right] + \mathbb{E}\left[u_{1i}\frac{\partial v_{1j}}{\partial W_{ijk}}w_{1k}\right] + \mathbb{E}\left[u_{1i}v_{1j}\frac{\partial w_{1k}}{\partial W_{ijk}}\right]$$

where the last equality is derived from Stein's Lemma and the involved derivatives express as

$$\frac{\partial u_{1i}}{\partial W_{ijk}} \simeq \frac{-1}{\sqrt{n}}v_{1j}w_{1k}R_{ii}^{11}(\lambda_1), \quad \frac{\partial v_{1j}}{\partial W_{ijk}} \simeq \frac{-1}{\sqrt{n}}u_{1i}w_{1k}R_{jj}^{22}(\lambda_1), \quad \frac{\partial w_{1k}}{\partial W_{ijk}} \simeq \frac{-1}{\sqrt{n}}u_{1i}v_{1j}R_{kk}^{33}(\lambda_1)$$

Substituting in the above sum, we get

$$\frac{1}{\sqrt{n}}\mathbb{E}\left[\mathcal{W}(u_1, v_1, w_1)\right] \simeq -\frac{1}{n}\sum_{ijk}\mathbb{E}[v_{1j}^2 w_{1k}^2 R_{ii}^{11}(\lambda_1)] - \frac{1}{n}\sum_{ijk}\mathbb{E}[u_{1i}^2 w_{1k}^2 R_{jj}^{22}(\lambda_1)] - \frac{1}{n}\sum_{ijk}\mathbb{E}[u_{1i}^2 v_{1j}^2 R_{kk}^{33}(\lambda_1)]$$

$$= -\mathbb{E}\left[\frac{1}{n}\operatorname{tr} R^{11}(\lambda_1)\right] - \mathbb{E}\left[\frac{1}{n}\operatorname{tr} R^{22}(\lambda_1)\right] - \mathbb{E}\left[\frac{1}{n}\operatorname{tr} R^{33}(\lambda_1)\right]$$

$$\xrightarrow[n\to\infty]{} -(r_1(\lambda_1) + r_2(\lambda_1) + r_3(\lambda_1)) = -r(\lambda_1)$$

Therefore, we have

$$\lambda_1 + r(\lambda_1) = \mathcal{S}(u_1, v_1, w_1) \tag{22}$$

### D.1.3 Estimation of the alignments

**Estimation of $\langle u_1, x_s \rangle$:** Again from the first identity in equation 16, we have

$$\lambda_1\langle u_1, x_s \rangle = \mathcal{T}_1(x_s, v_1, w_1) = \mathcal{S}(x_s, v_1, w_1) + \frac{1}{\sqrt{n}}\mathcal{W}(x_s, v_1, w_1)$$

And we have

$$\frac{1}{\sqrt{n}}\mathbb{E}[\mathcal{W}(x_s, v_1, w_1)] = \frac{1}{\sqrt{n}}\sum_{ijk}\mathbb{E}\left[x_{si}v_{1j}w_{1k}W_{ijk}\right]$$

$$= \frac{1}{\sqrt{n}}\sum_{ijk}\mathbb{E}\left[x_{si}\frac{\partial v_{1j}}{\partial W_{ijk}}w_{1k}\right] + \frac{1}{\sqrt{n}}\sum_{ijk}\mathbb{E}\left[x_{si}v_{1j}\frac{\partial w_{1k}}{\partial W_{ijk}}\right]$$

$$\simeq -\frac{1}{n}\sum_{ijk}\mathbb{E}\left[x_{si}u_{1i}w_{1k}^2 R_{jj}^{22}(\lambda_1)\right] - \frac{1}{n}\sum_{ijk}\mathbb{E}\left[x_{si}u_{1i}v_{1j}^2 R_{kk}^{33}(\lambda_1)\right]$$

$$= -\mathbb{E}\left[\langle x_s, u_1\rangle\frac{1}{n}\operatorname{tr} R^{22}(\lambda_1)\right] - \mathbb{E}\left[\langle x_s, u_1\rangle\frac{1}{n}\operatorname{tr} R^{33}(\lambda_1)\right]$$

$$\xrightarrow[n\to\infty]{} -(r_2(\lambda_1) + r_3(\lambda_1))\langle x_s, u_1\rangle = -(r(\lambda_1) - r_1(\lambda_1))\langle x_s, u_1\rangle$$

Therefore, we have

$$(\lambda_1 + r(\lambda_1) - r_1(\lambda_1))\langle x_s, u_1\rangle = \mathcal{S}(x_s, v_1, w_1) \tag{23}$$

Similarly, we get

$$(\lambda_1 + r(\lambda_1) - r_2(\lambda_1))\langle y_s, v_1\rangle = \mathcal{S}(u_1, y_s, w_1)$$
$$(\lambda_1 + r(\lambda_1) - r_3(\lambda_1))\langle z_s, w_1\rangle = \mathcal{S}(u_1, v_1, z_s) \tag{24}$$

Finally, with our assumption in equation 2 and since $\mathcal{T}_1$ is cubic, the above equations reduce to the following system of equations describing the first deflation step

$$\begin{cases} f_r(\lambda_1) = \sum_{i=1}^2 \beta_i \rho_{1i}^3 \\ h_r(\lambda_1)\rho_{1j} = \sum_{i=1}^2 \beta_i \langle x_i, x_j\rangle \rho_{1i}^2 \quad \text{for} \quad j \in [2] \end{cases} \tag{25}$$

where we denoted $f_r(z) = z + r(z)$ and $h_r(z) = z + \frac{2}{3}r(z) = -\frac{1}{r(z)}$.

## D.2 SECOND DEFLATION STEP

Given $u_1$ from the first deflation step, we consider now the following random tensor

$$\mathcal{T}_2 = \mathcal{T}_1 \times_1 \left(\boldsymbol{I}_N - \gamma u_1 u_1^\top\right) = \mathcal{T}_1 - \gamma u_1 \otimes \mathcal{T}_1(u_1, \cdot, \cdot) \tag{26}$$

Again the singular vectors of $\mathcal{T}_2$ satisfy

$$\mathcal{T}_2(\cdot, v_2, w_2) = \lambda_2 u_2, \quad \mathcal{T}_2(u_2, \cdot, w_2) = \lambda_2 v_2, \quad \mathcal{T}_2(u_2, v_2, \cdot) = \lambda_2 w_2 \tag{27}$$

and we also have

$$\begin{pmatrix} \frac{\partial u_2}{\partial W_{ijk}} \\ \frac{\partial v_2}{\partial W_{ijk}} \\ \frac{\partial w_2}{\partial W_{ijk}} \end{pmatrix} = -\frac{1}{\sqrt{n}}\left(\begin{pmatrix} 0 & \mathcal{T}_2(w_2) & \mathcal{T}_2(v_2) \\ \mathcal{T}_2(w_2)^\top & 0 & \mathcal{T}_2(u_2) \\ \mathcal{T}_2(v_2)^\top & \mathcal{T}_2(u_2)^\top & 0 \end{pmatrix} - \lambda_2\boldsymbol{I}_n\right)^{-1}\begin{pmatrix} v_{2j}w_{2k}(e_i - u_{2i}u_2) \\ u_{2i}w_{2k}(e_j - v_{2j}v_2) \\ u_{2i}v_{2j}(e_k - w_{2k}w_2) \end{pmatrix} \tag{28}$$

### D.2.1 STIELTJES TRANSFORM

Again, since $\mathcal{T}_1$ is a low-rank perturbation of a random tensor $\mathcal{W}$, it is easily noticed that

$$\begin{pmatrix} 0 & \mathcal{T}_2(w_2) & \mathcal{T}_2(v_2) \\ \mathcal{T}_2(w_2)^\top & 0 & \mathcal{T}_2(u_2) \\ \mathcal{T}_2(v_2)^\top & \mathcal{T}_2(u_2)^\top & 0 \end{pmatrix} = \boldsymbol{M} + \boldsymbol{P}$$

where $\boldsymbol{P}$ is some low-rank matrix and $\boldsymbol{M}$ is a random matrix given by

$$\boldsymbol{M} = \frac{1}{\sqrt{n}}\begin{pmatrix} 0 & \mathcal{W}(w_2) & \mathcal{W}(v_2) \\ \mathcal{W}(w_2)^\top & 0 & \mathcal{W}(u_2) - \gamma\langle u_1, u_2\rangle\mathcal{W}(u_1) \\ \mathcal{W}(v_2)^\top & \mathcal{W}(u_2)^\top - \gamma\langle u_1, u_2\rangle\mathcal{W}(u_1)^\top & 0 \end{pmatrix} \tag{29}$$

Therefore, by Lemma C.2, the limiting Stieltjes transform corresponding to the analysis of the second deflation step can be computed through the resolvent $Q(z) = (M - zI_n)^{-1}$ of the random matrix $M$ and we denote $\kappa = \langle u_1, u_2 \rangle$. We denote the different sub-blocks of $Q(z)$ as

$$Q(z) = \begin{pmatrix} Q^{11}(z) & Q^{12}(z) & Q^{13}(z) \\ Q^{12}(z)^\top & Q^{22}(z) & Q^{23}(z) \\ Q^{13}(z)^\top & Q^{23}(z)^\top & Q^{33}(z) \end{pmatrix} \tag{30}$$

Denote

$$\frac{1}{n} \operatorname{tr} Q^{ii}(z) \xrightarrow[n\to\infty]{} q_i(z) \quad \text{and} \quad \frac{1}{n} \operatorname{tr} Q(z) \xrightarrow[n\to\infty]{} q(z) \tag{31}$$

**Estimation of $\frac{1}{n} \operatorname{tr} Q^{11}(z)$:** From the identity $MQ(z) - zQ(z) = I_n$, we have

$$\frac{1}{\sqrt{n}} \left[ \mathcal{W}(w_2)(Q^{12})^\top \right]_{ii} + \frac{1}{\sqrt{n}} \left[ \mathcal{W}(v_2)(Q^{13})^\top \right]_{ii} - zQ^{11}_{ii} = 1$$

Therefore

$$\frac{1}{n\sqrt{n}} \sum_{ijk} \mathbb{E} \left[ w_{2k} W_{ijk} Q^{12}_{ij} \right] + \frac{1}{n\sqrt{n}} \sum_{ijk} \mathbb{E} \left[ v_{2j} W_{ijk} Q^{13}_{ik} \right] - \frac{z}{n} \operatorname{tr} Q^{11}(z) = \frac{1}{3}$$

where

- $$\frac{1}{n\sqrt{n}} \sum_{ijk} \mathbb{E} \left[ w_{2k} W_{ijk} Q^{12}_{ij} \right] \simeq \frac{1}{n\sqrt{n}} \sum_{ijk} \mathbb{E} \left[ w_{2k} \frac{\partial Q^{12}_{ij}}{\partial W_{ijk}} \right]$$

From Seddik et al. (2021), we have $\frac{\partial Q^{12}_{ij}}{\partial W_{ijk}} \simeq -\frac{1}{\sqrt{n}} w_{2k} Q^{11}_{ii} Q^{22}_{jj}$, hence

$$\frac{1}{n\sqrt{n}} \sum_{ijk} \mathbb{E} \left[ w_{2k} W_{ijk} Q^{12}_{ij} \right] \simeq -\frac{1}{n^2} \sum_{ijk} \mathbb{E} \left[ w^2_{2k} Q^{11}_{ii} Q^{22}_{jj} \right] = -\mathbb{E} \left[ \frac{1}{n} \operatorname{tr} Q^{11} \frac{1}{n} \operatorname{tr} Q^{22} \right] \xrightarrow[n\to\infty]{} -q_1(z)q_2(z)$$

Similarly, we have

- $$\frac{1}{n\sqrt{n}} \sum_{ijk} \mathbb{E} \left[ v_{2j} W_{ijk} Q^{13}_{ik} \right] \xrightarrow[n\to\infty]{} -q_1(z)q_3(z)$$

Therefore, $q_1(z) = \lim_{n\to\infty} \frac{1}{n} \operatorname{tr} Q^{11}(z)$ satisfies the equation

$$[q_2(z) + q_3(z) + z]q_1(z) + \frac{1}{3} = 0 \tag{32}$$

**Estimation of $\frac{1}{n} \operatorname{tr} Q^{22}(z)$:** We have

$$\frac{1}{\sqrt{n}} \left[ \mathcal{W}(w_2)^\top Q^{12} \right]_{jj} + \frac{1}{\sqrt{n}} \left[ (\mathcal{W}(u_2) - \gamma\kappa\mathcal{W}(u_1))(Q^{23})^\top \right]_{jj} - zQ^{22}_{jj} = 1$$

Hence

$$\frac{1}{n\sqrt{n}} \sum_{ijk} \mathbb{E} \left[ w_{2k} W_{ijk} Q^{12}_{ij} \right] + \frac{1}{n\sqrt{n}} \sum_{ijk} \mathbb{E} \left[ u_{2i} W_{ijk} Q^{23}_{jk} \right] - \frac{\gamma\kappa}{n\sqrt{n}} \sum_{ijk} \mathbb{E} \left[ u_{1i} W_{ijk} Q^{23}_{jk} \right] - \frac{z}{n} \operatorname{tr} Q^{22} = \frac{1}{3}$$

where

- $$\frac{1}{n\sqrt{n}} \sum_{ijk} \mathbb{E} \left[ w_{2k} W_{ijk} Q^{12}_{ij} \right] \simeq \frac{1}{n\sqrt{n}} \sum_{ijk} \mathbb{E} \left[ w_{2k} \frac{\partial Q^{12}_{ij}}{\partial W_{ijk}} \right] = -\frac{1}{n^2} \sum_{ijk} \mathbb{E} \left[ w^2_{2k} Q^{11}_{ii} Q^{22}_{jj} \right] \xrightarrow[n\to\infty]{} -q_1(z)q_2(z)$$

- $$\frac{1}{n\sqrt{n}} \sum_{ijk} \mathbb{E} \left[ u_{2i} W_{ijk} Q^{23}_{jk} \right] \simeq \frac{1}{n\sqrt{n}} \sum_{ijk} \mathbb{E} \left[ u_{2i} \frac{\partial Q^{23}_{jk}}{\partial W_{ijk}} \right] = -\frac{1}{n^2} \sum_{ijk} \mathbb{E} \left[ (u^2_{2i} - \gamma\kappa u_{1i} u_{2i}) Q^{22}_{jj} Q^{33}_{kk} \right]$$

$$\xrightarrow[n\to\infty]{} (\gamma\kappa^2 - 1)q_2(z)q_3(z)$$

- $\dfrac{1}{n\sqrt{n}}\sum_{ijk}\mathbb{E}\left[u_{1i}W_{ijk}Q^{23}_{jk}\right] \simeq \dfrac{1}{n\sqrt{n}}\sum_{ijk}\mathbb{E}\left[u_{1i}\dfrac{\partial Q^{23}_{jk}}{\partial W_{ijk}}\right] = -\dfrac{1}{n^2}\sum_{ijk}\mathbb{E}\left[(u_{1i}u_{2i}-\gamma\kappa u^2_{1i})Q^{22}_{jj}Q^{33}_{kk}\right]$

$$\xrightarrow[n\to\infty]{} \kappa(\gamma-1)q_2(z)q_3(z)$$

where we used the fact that $\dfrac{\partial Q^{23}_{jk}}{\partial W_{ijk}} \simeq -\dfrac{1}{\sqrt{n}}(u_{2i}-\gamma\kappa u_{1i})Q^{22}_{jj}Q^{33}_{kk}$.

$$\left(q_1(z)+z-\left[\gamma\kappa^2-1+\kappa(\gamma-1)\right]q_3(z)\right)q_2(z)+\frac{1}{3}=0 \tag{33}$$

**Estimation of $\frac{1}{n}\operatorname{tr}Q^{33}(z)$:** From the identity $MQ(z)-zQ(z)=I_n$, we have

$$\frac{1}{\sqrt{n}}\left[\mathcal{W}(v_2)^\top Q^{13}\right]_{kk} + \frac{1}{\sqrt{n}}\left[\left(\mathcal{W}(u_2)^\top-\gamma\kappa\mathcal{W}(u_1)^\top\right)(Q^{23})\right]_{kk} - zQ^{33}_{kk}=1$$

Hence

$$\frac{1}{n\sqrt{n}}\sum_{ijk}\mathbb{E}\left[v_{2j}W_{ijk}Q^{13}_{ik}\right] + \frac{1}{n\sqrt{n}}\sum_{ijk}\mathbb{E}\left[u_{2i}W_{ijk}Q^{23}_{jk}\right] - \gamma\kappa\frac{1}{n\sqrt{n}}\sum_{ijk}\mathbb{E}\left[u_{1i}W_{ijk}Q^{23}_{jk}\right] - \frac{z}{n}\operatorname{tr}Q^{33}(z)=\frac{1}{3}$$

where

$$\frac{1}{n\sqrt{n}}\sum_{ijk}\mathbb{E}\left[v_{2j}W_{ijk}Q^{13}_{ik}\right] \simeq \frac{1}{n\sqrt{n}}\sum_{ijk}\mathbb{E}\left[v_{2j}\frac{\partial Q^{13}_{ik}}{\partial W_{ijk}}\right] = -\frac{1}{n^2}\sum_{ijk}\mathbb{E}\left[v^2_{2j}Q^{11}_{ii}Q^{33}_{kk}\right] \xrightarrow[n\to\infty]{} -q_1(z)q_3(z)$$

$$\frac{1}{n\sqrt{n}}\sum_{ijk}\mathbb{E}\left[u_{2i}W_{ijk}Q^{23}_{jk}\right] \simeq \frac{1}{n\sqrt{n}}\sum_{ijk}\mathbb{E}\left[u_{2i}\frac{\partial Q^{23}_{jk}}{\partial W_{ijk}}\right] = -\frac{1}{n^2}\sum_{ijk}\mathbb{E}\left[\left(u^2_{2i}-\gamma\kappa u_{1i}u_{2i}\right)Q^{22}_{jj}Q^{33}_{kk}\right]$$

$$\xrightarrow[n\to\infty]{} \left(\gamma\kappa^2-1\right)q_2(z)q_3(z)$$

$$\frac{1}{n\sqrt{n}}\sum_{ijk}\mathbb{E}\left[u_{1i}W_{ijk}Q^{23}_{jk}\right] \simeq \frac{1}{n\sqrt{n}}\sum_{ijk}\mathbb{E}\left[u_{1i}\frac{\partial Q^{23}_{jk}}{\partial W_{ijk}}\right] = -\frac{1}{n^2}\sum_{ijk}\mathbb{E}\left[(u_{1i}u_{2i}-\gamma\kappa u^2_{1i})Q^{22}_{jj}Q^{33}_{kk}\right]$$

$$\xrightarrow[n\to\infty]{} \kappa(\gamma-1)q_2(z)q_3(z)$$

with again $\dfrac{\partial Q^{23}_{jk}}{\partial W_{ijk}} \simeq -\dfrac{1}{\sqrt{n}}(u_{2i}-\gamma\kappa u_{1i})Q^{22}_{jj}Q^{33}_{kk}$.

$$\left(q_1(z)+z-\left[\gamma\kappa^2-1+\kappa(\gamma-1)\right]q_2(z)\right)q_3(z)+\frac{1}{3}=0 \tag{34}$$

Therefore, we have

$$\begin{cases} [q_2(z)+q_3(z)+z]q_1(z)+\frac{1}{3}=0 \\ \left(q_1(z)+z-\left[\gamma\kappa^2-1+\kappa(\gamma-1)\right]q_3(z)\right)q_2(z)+\frac{1}{3}=0 \\ \left(q_1(z)+z-\left[\gamma\kappa^2-1+\kappa(\gamma-1)\right]q_2(z)\right)q_3(z)+\frac{1}{3}=0 \\ q(z)=\sum_{i=1}^{3}q_i(z) \end{cases} \tag{35}$$

Moreover, by symmetry in equation 2 and since $\mathcal{T}_1$ is cubic, we have $b(z)=q_2(z)=q_3(z)$ and we denote $a(z)=q_1(z)$ and $\tau=\gamma\kappa^2-1+\kappa(\gamma-1)$. Hence,

$$\begin{cases} [2b(z)+z]\,a(z)+\frac{1}{3}=0 \\ (a(z)+z-\tau b(z))b(z)+\frac{1}{3}=0 \end{cases} \tag{36}$$

Moreover, $q(z)=a(z)+2b(z)$.

### D.2.2 ESTIMATION OF THE SINGULAR VALUE

**Estimation of $\lambda_2$:** We first have

$$\lambda_2 = \mathcal{T}_2(u_2, v_2, w_2) = \mathcal{T}_1(u_2, v_2, w_2) - \gamma\langle u_1, u_2\rangle\mathcal{T}_1(u_1, v_2, w_2)$$

$$= \mathcal{S}(u_2, v_2, w_2) + \frac{1}{\sqrt{n}}\mathcal{W}(u_2, v_2, w_2) - \gamma\langle u_1, u_2\rangle\left(\mathcal{S}(u_1, v_2, w_2) + \frac{1}{\sqrt{n}}\mathcal{W}(u_1, v_2, w_2)\right)$$

where we have

$$\frac{1}{\sqrt{n}}\mathbb{E}\left[\mathcal{W}(u_2, v_2, w_2)\right] \xrightarrow[n\to\infty]{} -q(\lambda_2)$$

and

$$\frac{1}{\sqrt{n}}\mathbb{E}\left[\mathcal{W}(u_1, v_2, w_2)\right] = \frac{1}{\sqrt{n}}\sum_{ijk}\mathbb{E}\left[u_{1i}v_{2j}w_{2k}W_{ijk}\right]$$

$$= \frac{1}{\sqrt{n}}\sum_{ijk}\mathbb{E}\left[\frac{\partial u_{1i}}{\partial W_{ijk}}v_{2j}w_{2k}\right] + \mathbb{E}\left[u_{1i}\frac{\partial v_{2j}}{\partial W_{ijk}}w_{2k}\right] + \mathbb{E}\left[u_{1i}v_{2j}\frac{\partial w_{2k}}{\partial W_{ijk}}\right]$$

Again, we have

$$\frac{\partial u_{2i}}{\partial W_{ijk}} \simeq \frac{-1}{\sqrt{n}}v_{2j}w_{2k}Q_{ii}^{11}(\lambda_2), \quad \frac{\partial v_{2j}}{\partial W_{ijk}} \simeq \frac{-1}{\sqrt{n}}u_{2i}w_{2k}Q_{jj}^{22}(\lambda_2), \quad \frac{\partial w_{2k}}{\partial W_{ijk}} \simeq \frac{-1}{\sqrt{n}}u_{2i}v_{2j}Q_{kk}^{33}(\lambda_2)$$

Therefore

$$\frac{1}{\sqrt{n}}\mathbb{E}\left[\mathcal{W}(u_1, v_2, w_2)\right] \simeq -\frac{1}{n}\sum_{ijk}\mathbb{E}\left[v_{1j}w_{1k}v_{2j}w_{2k}R_{ii}^{11}(\lambda_1)\right] - \frac{1}{n}\sum_{ijk}\mathbb{E}\left[u_{1i}u_{2i}w_{2k}^2Q_{jj}^{22}(\lambda_2)\right]$$

$$- \frac{1}{n}\sum_{ijk}\mathbb{E}\left[u_{1i}v_{2j}^2u_{2i}Q_{kk}^{33}(\lambda_2)\right]$$

$$\xrightarrow[n\to\infty]{} -\langle v_1, v_2\rangle\langle w_1, w_2\rangle r_1(\lambda_1) - \langle u_1, u_2\rangle q_2(\lambda_2) - \langle u_1, u_2\rangle q_3(\lambda_2)$$

$$= -\langle v_1, v_2\rangle\langle w_1, w_2\rangle r_1(\lambda_1) - \langle u_1, u_2\rangle\left[q_2(\lambda_2) + q_3(\lambda_2)\right]$$

Hence, $\lambda_2$ satisfies

$$\begin{aligned}\lambda_2 + q(\lambda_2) - \gamma\langle u_1, u_2\rangle\langle v_1, v_2\rangle\langle w_1, w_2\rangle r_1(\lambda_1) - \gamma\langle u_1, u_2\rangle^2\left[q_2(\lambda_2) + q_3(\lambda_2)\right] \\ = \mathcal{S}(u_2, v_2, w_2) - \gamma\langle u_1, u_2\rangle\mathcal{S}(u_1, v_2, w_2)\end{aligned} \tag{37}$$

And by symmetry, from equation 2 and since $\mathcal{T}_1$ is cubic, we have

$$\boxed{f_q(\lambda_2) - \frac{\gamma\kappa\eta^2}{3}r(\lambda_1) - 2\gamma\kappa^2 b(\lambda_2) = \sum_{i=1}^{2}\beta_i\theta_{2i}\rho_{2i}^2 - \gamma\kappa\sum_{i=1}^{2}\beta_i\rho_{1i}\rho_{2i}^2} \tag{38}$$

where we denoted $f_q(z) = z + q(z)$ and

$$\theta_{2i} = \langle u_2, x_i\rangle, \quad \rho_{2i} = \langle v_2, y_i\rangle = \langle w_2, z_i\rangle, \quad \kappa = \langle u_1, u_2\rangle, \quad \eta = \langle v_1, v_2\rangle = \langle w_1, w_2\rangle$$

### D.2.3 ESTIMATION OF THE ALIGNMENTS

**Estimation of $\langle u_2, x_s\rangle$:** From the identity in equation 27, we have

$$\lambda_2\langle u_2, x_s\rangle = \mathcal{T}_2(x_s, v_2, w_2) = \mathcal{T}_1(x_s, v_2, w_2) - \gamma\langle u_1, x_s\rangle\mathcal{T}_1(u_1, v_2, w_2)$$

$$= \mathcal{S}(x_s, v_2, w_2) + \frac{1}{\sqrt{n}}\mathcal{W}(x_s, v_2, w_2) - \gamma\langle u_1, x_s\rangle\left(\mathcal{S}(u_1, v_2, w_2) + \frac{1}{\sqrt{n}}\mathcal{W}(u_1, v_2, w_2)\right)$$

where

$$\frac{1}{\sqrt{n}}\mathbb{E}\left[\mathcal{W}(x_s, v_2, w_2)\right] \xrightarrow[n\to\infty]{} -(q(\lambda_2) - q_1(\lambda_2))\langle x_s, u_2\rangle$$

and $\mathbb{E}\left[\frac{1}{\sqrt{n}}\mathcal{W}(u_1, v_2, w_2)\right]$ was computed previously. We therefore have

$$[\lambda_2 + q(\lambda_2) - q_1(\lambda_2)]\langle x_s, u_2 \rangle - \gamma\langle u_1, x_s \rangle [\langle v_1, v_2 \rangle \langle w_1, w_2 \rangle r_1(\lambda_1) + \langle u_1, u_2 \rangle (q_2(\lambda_2) + q_3(\lambda_2))]$$
$$= \mathcal{S}(x_s, v_2, w_2) - \gamma\langle x_s, u_1 \rangle \mathcal{S}(u_1, v_2, w_2)$$

(39)

Again by symmetry, from equation 2 and since $\mathcal{T}_1$ is cubic, we have

$$\boxed{[f_q(\lambda_2) - a(\lambda_2)]\theta_{2s} - \gamma\rho_{1s}\left[\frac{\eta^2}{3}r(\lambda_1) + 2\kappa b(\lambda_2)\right] = \sum_{i=1}^{2}\beta_i\langle x_s, x_i \rangle\rho_{2i}^2 - \gamma\rho_{1s}\sum_{i=1}^{2}\beta_i\rho_{1i}\rho_{2i}^2 \quad \text{for} \quad s \in [2]}$$

(40)

**Estimation of $\langle u_1, u_2 \rangle$:** Again from equation 27, we have

$$\lambda_2\langle u_1, u_2 \rangle = \mathcal{T}_2(u_1, v_2, v_2)$$

with $\mathcal{T}_2 = \mathcal{T}_1 - \gamma u_1 \otimes \mathcal{T}_1(u_1, \cdot, \cdot)$, therefore

$$\lambda_2\langle u_1, u_2 \rangle = \mathcal{T}_1(u_1, v_2, w_2) - \gamma\mathcal{T}_1(u_1, v_2, w_2) = (1 - \gamma)\mathcal{T}_1(u_1, v_2, w_2)$$
$$= (1 - \gamma)\left(\mathcal{S}(u_1, v_2, w_2) + \frac{1}{\sqrt{n}}\mathcal{W}(u_1, v_2, w_2)\right)$$

(41)

Hence, we have

$$[\lambda_2 + (1 - \gamma)(q_2(\lambda_2) + q_3(\lambda_2))]\langle u_1, u_2 \rangle = (1 - \gamma)[\mathcal{S}(u_1, v_2, w_2) - \langle v_1, v_2 \rangle\langle w_1, w_2 \rangle r_1(\lambda_1)]$$

(42)

Again by symmetry, from equation 2 and since $\mathcal{T}_1$ is cubic, we have

$$\boxed{[\lambda_2 + 2(1 - \gamma)b(\lambda_2)]\kappa = (1 - \gamma)\left[\sum_{i=1}^{2}\beta_i\rho_{1i}\rho_{2i}^2 - \frac{\eta^2}{3}r(\lambda_1)\right]}$$

(43)

**Estimation of $\langle v_2, y_s \rangle$:** From equation 27, we have

$$\lambda_2\langle v_2, y_s \rangle = \mathcal{T}_2(u_2, y_s, w_2) = \mathcal{T}_1(u_2, y_s, w_2) - \gamma\langle u_1, u_2 \rangle\mathcal{T}_1(u_1, y_s, w_2)$$
$$= \mathcal{S}(u_2, y_s, w_2) + \frac{1}{\sqrt{n}}\mathcal{W}(u_2, y_s, w_2) - \gamma\langle u_1, u_2 \rangle\left[\mathcal{S}(u_1, y_s, w_2) + \frac{1}{\sqrt{n}}\mathcal{W}(u_1, y_s, w_2)\right]$$

And as previously, we have

$$\mathbb{E}\left[\frac{1}{\sqrt{n}}\mathcal{W}(u_2, y_s, w_2)\right] \xrightarrow[n \to \infty]{} -(q(\lambda_2) - q_2(\lambda_2))\langle y_s, v_2 \rangle$$

And

$$\mathbb{E}\left[\frac{1}{\sqrt{n}}\mathcal{W}(u_1, y_s, w_2)\right] = \frac{1}{\sqrt{n}}\sum_{ijk}y_{sj}\mathbb{E}\left[\frac{\partial u_{1i}}{\partial W_{ijk}}w_{2k} + u_{1i}\frac{\partial w_{2k}}{\partial W_{ijk}}\right]$$

$$\simeq -\frac{1}{n}\sum_{ijk}y_{sj}\mathbb{E}\left[v_{1j}w_{1k}R_{ii}^{11}(\lambda_1)w_{2k} + u_{1i}u_{2i}v_{2j}Q_{kk}^{33}(\lambda_2)\right]$$

$$\xrightarrow[n \to \infty]{} -\langle y_s, v_1 \rangle\langle w_1, w_2 \rangle r_1(\lambda_1) - \langle y_s, v_2 \rangle\langle u_1, u_2 \rangle q_3(\lambda_2)$$

Therefore,

$$\left[\lambda_2 + q(\lambda_2) - q_2(\lambda_2) - \gamma\langle u_1, u_2 \rangle^2 q_3(\lambda_2)\right]\langle v_2, y_s \rangle =$$
$$\mathcal{S}(u_2, y_s, w_2) - \gamma\langle u_1, u_2 \rangle[\mathcal{S}(u_1, y_s, w_2) - \langle v_1, y_s \rangle\langle w_1, w_2 \rangle r_1(\lambda_1)]$$

(44)

Again by symmetry, from equation 2 and since $\mathcal{T}_1$ is cubic, we have

$$\boxed{[f_q(\lambda_2) - (1 + \gamma\kappa^2)b(\lambda_2)]\rho_{2s} = \sum_{i=1}^{2}\beta_i\theta_{2i}\rho_{2i}\langle y_s, y_i \rangle - \gamma\kappa\left[\sum_{i=1}^{2}\beta_i\rho_{1i}\rho_{2i}\langle y_s, y_i \rangle - \frac{\rho_{1s}\eta}{3}r(\lambda_1)\right] \quad \text{for} \quad s \in [2]}$$

(45)

**Estimation of $\langle v_1, v_2 \rangle$:** From equation 27, we have

$$\lambda_2 \langle v_1, v_2 \rangle = \mathcal{T}_2(u_2, v_1, w_2) = \mathcal{T}_1(u_2, v_1, w_2) - \gamma \langle u_1, u_2 \rangle \mathcal{T}_1(u_1, v_1, w_2)$$

$$= \mathcal{S}(u_2, v_1, w_2) + \frac{1}{\sqrt{n}} \mathcal{W}(u_2, v_1, w_2) - \gamma \langle u_1, u_2 \rangle \left[ \mathcal{S}(u_1, v_1, w_2) + \frac{1}{\sqrt{n}} \mathcal{W}(u_1, v_1, w_2) \right]$$

Where

$$\mathbb{E}\left[ \frac{1}{\sqrt{n}} \mathcal{W}(u_2, v_1, w_2) \right] = \frac{1}{\sqrt{n}} \sum_{ijk} \mathbb{E}[u_{2i} v_{1j} w_{2k} W_{ijk}]$$

$$= \frac{1}{\sqrt{n}} \sum_{ijk} \mathbb{E}\left[ \frac{\partial u_{2i}}{\partial W_{ijk}} v_{1j} w_{2k} + u_{2i} \frac{\partial v_{1j}}{\partial W_{ijk}} w_{2k} + u_{2i} v_{1j} \frac{\partial w_{2k}}{\partial W_{ijk}} \right]$$

$$\simeq -\frac{1}{n} \sum_{ijk} \mathbb{E}\left[ v_{2j} w_{2k} Q_{ii}^{11}(\lambda_2) v_{1j} w_{2k} + u_{2i} u_{1i} w_{1k} R_{jj}^{22}(\lambda_1) w_{2k} + u_{2i} v_{1j} u_{2i} v_{2j} Q_{kk}^{33}(\lambda_2) \right]$$

$$\xrightarrow[n \to \infty]{} -\langle v_1, v_2 \rangle [q_1(\lambda_2) + q_3(\lambda_2)] - \langle u_1, u_2 \rangle \langle w_1, w_2 \rangle r_2(\lambda_1)$$

And

$$\mathbb{E}\left[ \frac{1}{\sqrt{n}} \mathcal{W}(u_1, v_1, w_2) \right] = \frac{1}{\sqrt{n}} \sum_{ijk} \mathbb{E}[u_{1i} v_{1j} w_{2k} W_{ijk}]$$

$$= \frac{1}{\sqrt{n}} \sum_{ijk} \mathbb{E}\left[ \frac{\partial u_{1i}}{\partial W_{ijk}} v_{1j} w_{2k} + u_{1i} \frac{\partial v_{1j}}{\partial W_{ijk}} w_{2k} + u_{1i} v_{1j} \frac{\partial w_{2k}}{\partial W_{ijk}} \right]$$

$$\simeq -\frac{1}{n} \sum_{ijk} \mathbb{E}\left[ v_{1j} w_{1k} R_{ii}^{11}(\lambda_1) v_{1j} w_{2k} + u_{1i}^2 w_{1k} w_{2k} R_{jj}^{22}(\lambda_1) + u_{1i} v_{1j} u_{2i} v_{2j} Q_{kk}^{33}(\lambda_2) \right]$$

$$\xrightarrow[n \to \infty]{} -\langle w_1, w_2 \rangle [r_1(\lambda_1) + r_2(\lambda_1)] - \langle u_1, u_2 \rangle \langle v_1, v_2 \rangle q_3(\lambda_2)$$

Hence, we have

$$\left[ \lambda_2 + q_1(\lambda_2) + q_3(\lambda_2) - \gamma \langle u_1, u_2 \rangle^2 q_3(\lambda_2) \right] \langle v_1, v_2 \rangle + \left[ (1 - \gamma) r_2(\lambda_1) - r_1(\lambda_1) \right] \langle u_1, u_2 \rangle \langle w_1, w_2 \rangle$$
$$= \mathcal{S}(u_2, v_1, w_2) - \gamma \langle u_1, u_2 \rangle \mathcal{S}(u_1, v_1, w_2)$$

$$(46)$$

Finally by symmetry, from equation 2 and since $\mathcal{T}_1$ is cubic, we have

$$\boxed{\left[ \lambda_2 + a(\lambda_2) + (1 - \gamma \kappa^2) b(\lambda_2) - \frac{\gamma \kappa}{3} r(\lambda_1) \right] \eta = \sum_{i=1}^{2} \beta_i \theta_{2i} \rho_{1i} \rho_{2i} - \gamma \kappa \sum_{i=1}^{2} \beta_i \rho_{1i}^2 \rho_{2i}} \qquad (47)$$

### D.2.4  $\gamma$-ORTHOGONALIZED TENSOR DEFLATION: GOVERNING SYSTEM OF EQUATIONS

The second deflation step is therefore governed by the following system of equations

$$\begin{cases} [2b(z) + z] a(z) + \frac{1}{3} = 0 \\ (a(z) + z - \tau b(z)) b(z) + \frac{1}{3} = 0 \\ q(z) = a(z) + 2b(z) \\ f_q(\lambda_2) - \frac{\gamma \kappa \eta^2}{3} r(\lambda_1) - 2\gamma \kappa^2 b(\lambda_2) = \sum_{i=1}^{2} \beta_i \theta_{2i} \rho_{2i}^2 - \gamma \kappa \sum_{i=1}^{2} \beta_i \rho_{1i} \rho_{2i}^2 \\ [f_q(\lambda_2) - a(\lambda_2)] \theta_{2s} - \gamma \rho_{1s} \left[ \frac{\eta^2}{3} r(\lambda_1) + 2\kappa b(\lambda_2) \right] = \sum_{i=1}^{2} \beta_i \langle x_s, x_i \rangle \rho_{2i}^2 - \gamma \rho_{1s} \sum_{i=1}^{2} \beta_i \rho_{1i} \rho_{2i}^2 \\ [\lambda_2 + 2(1 - \gamma) b(\lambda_2)] \kappa = (1 - \gamma) \left[ \sum_{i=1}^{2} \beta_i \rho_{1i} \rho_{2i}^2 - \frac{\eta^2}{3} r(\lambda_1) \right] \\ [f_q(\lambda_2) - (1 + \gamma \kappa^2) b(\lambda_2)] \rho_{2s} = \sum_{i=1}^{2} \beta_i \theta_{2i} \rho_{2i} \langle y_s, y_i \rangle - \gamma \kappa \left[ \sum_{i=1}^{2} \beta_i \rho_{1i} \rho_{2i} \langle y_s, y_i \rangle - \frac{\rho_{1s} \eta}{3} r(\lambda_1) \right] \\ [\lambda_2 + a(\lambda_2) + (1 - \gamma \kappa^2) b(\lambda_2) - \frac{\gamma \kappa}{3} r(\lambda_1)] \eta = \sum_{i=1}^{2} \beta_i \theta_{2i} \rho_{1i} \rho_{2i} - \gamma \kappa \sum_{i=1}^{2} \beta_i \rho_{1i}^2 \rho_{2i} \end{cases}$$

$$(48)$$

with $f_q(z) = z + q(z)$ and $\tau = \gamma\kappa^2 - 1 + \kappa(\gamma - 1)$. In the case $\gamma = 1$, we have $\kappa = 0$ from equation 41 and therefore the system above reduces to the following system, since $a(z) = b(z) = \frac{r(z)}{3}$ and $q(z) = r(z)$.

$$
\begin{cases}
f_r(\lambda_2) = \sum_{i=1}^{2} \beta_i \theta_{2i} \rho_{2i}^2 \\
h_r(\lambda_2)\theta_{2s} - \frac{\eta^2}{3} r(\lambda_1)\rho_{1s} = \sum_{i=1}^{2} \beta_i \langle x_s, x_i \rangle \rho_{2i}^2 - \rho_{1s} \sum_{i=1}^{2} \beta_i \rho_{1i} \rho_{2i}^2 \\
h_r(\lambda_2)\rho_{2s} = \sum_{i=1}^{2} \beta_i \theta_{2i} \rho_{2i} \langle y_s, y_i \rangle \\
\left[\lambda_2 + \frac{2}{3} r(\lambda_2)\right] \eta = \sum_{i=1}^{2} \beta_i \theta_{2i} \rho_{1i} \rho_{2i}
\end{cases}
\tag{49}
$$

# E  EXPERIMENTAL SETUP & SUPPLEMENTARY SIMULATIONS

## E.1  EXPERIMENTAL SETUP

We outline how we can generate synthetic models corresponding to our setting (2.1) in the following code snippet.

```python
import numpy as np
from numpy.random import randn, normal

from tensorly.tenalg import outer

class TwoSpikesModel():
    def __init__(self, b1, b2, alpha, dims):
        """
        Generates a rank-2 order-3 spiked tensor model as in Section 2.

        Parameters
        ----------
        b1, b2: Signal to Noise Ratios (SNRs)
        alpha: Correlation level between the 2 signal components.
        dims: Dimensions per mode up to order.
        ----------
        """
        self.b1 = b1
        self.b2 = b2
        self.order = len(dims)
        self.alpha = alpha
        self.c1 = [normalized(randn(d)) for d in dims]
        self.c1_orth = [(np.eye(d) - np.outer(u, u)) @ normalized(randn(d
                       )) for (d, u) in zip(dims, self.c1)]
        self.c2 = [alpha * u1 + np.sqrt(1 - alpha**2) * u2 for (u1, u2)
                  in zip(self.c1, self.c1_orth)]
        self.noise = normal(size=dims) / np.sqrt(sum(dims))
        self.tensor = b1 * outer(self.c1) + b2 * outer(self.c2) + self.
                     noise
```

## E.2  JOINT ESTIMATION OF MODEL PARAMETERS

In a variety of applications, we do not only care about estimating the hidden low rank components, but we also would like to estimate the different problem parameters with a sufficiently high accuracy. For instance, in telecommunications, a very fundamental problem is estimating the signal-to-noise strengths (SNRs) Pauluzzi & Beaulieu (2000); Wiesel et al. (2006); Suhadi et al. (2010); Matzner & Englberger (1994), which corresponds to quantifying the communication channel quality. In that spirit, we also address the problem of estimating the underlying model parameters, namely the signal-to-noise ratios $\beta_1, \beta_2$, and the correlation parameter $\alpha$ based on a **single realization** of $\mathcal{T}_1$. This allows us to design an improved deflation algorithm in the presence of correlations among our hidden signal components.

In this section, we discuss the problem of estimating the underlying model parameters, namely the SNRs and the signal components correlation $\boldsymbol{\beta} \equiv (\beta_1, \beta_2, \alpha) \in \mathbb{R}^3$, and the alignments $\boldsymbol{\rho} \equiv (\rho_{1i}, \rho_{2i}, \theta_{2i} \mid i \in [2]) \in \mathbb{R}^6$ from one realization of the random tensor $\mathcal{T}_1$. Indeed, this will allow us to design an improved deflation algorithm by optimizing the parameter $\gamma$ introduced

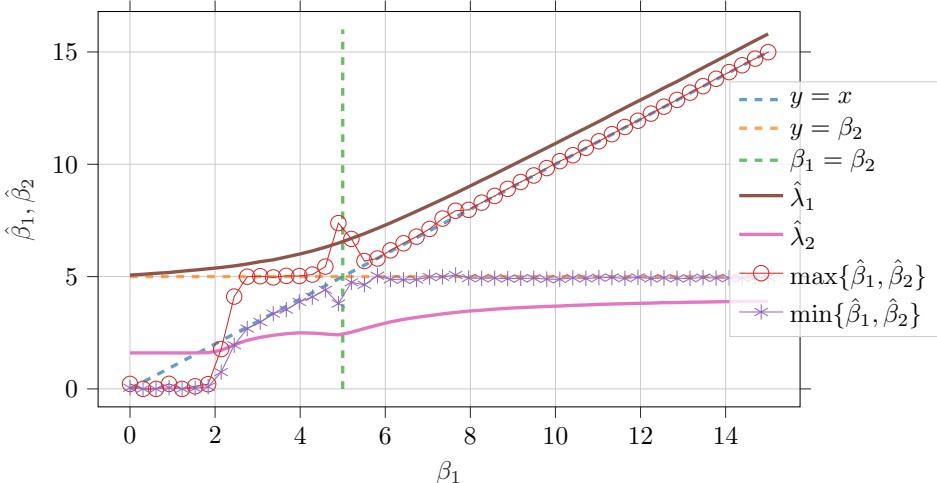

Figure 6: Estimation of the **underlying SNRs** $\beta_1$ and $\beta_2$ as described in Section 3. (Averaged over) 100 realizations of with $\mathcal{T}_1$ Parameters: $\beta_2 = 5$, $\alpha = 0.5$, $p = 150$ and $\gamma = 1$, $\beta_1$ is varied. The parameters are estimated only from the singular values $\hat{\lambda}_1, \hat{\lambda}_2$ and the alignment between the singular vectors $\hat{\eta} = \langle \hat{v}_1, \hat{v}_2 \rangle$, computed via tensor power iteration.

in the second deflation step. Further denoting $\boldsymbol{\lambda} \equiv (\lambda_1, \lambda_2, \eta) \in \mathbb{R}^3$, we define the mapping $\psi : \mathbb{R}^3 \times \mathbb{R}^3 \times \mathbb{R}^6 \to \mathbb{R}^9$ through equation 50, where the first three entries of the vector $\psi(\boldsymbol{\beta}, \boldsymbol{\lambda}, \boldsymbol{\rho})$ correspond to the first deflation step equations in equation 8 while the remaining entries correspond to the second deflation step for $\gamma = 1$ characterized by equation 51. In particular, the singular values $\lambda_1, \lambda_2$ and the corresponding alignments satisfy $\psi(\boldsymbol{\beta}, \boldsymbol{\lambda}, \boldsymbol{\rho}) = 0$. On the other hand, given an estimate $\hat{\boldsymbol{\lambda}} = (\hat{\lambda}_1, \hat{\lambda}_2, \hat{\eta})$ of $\boldsymbol{\lambda}$, which can be computed via tensor power iteration as discussed previously, we can solve $\psi(\cdot, \hat{\boldsymbol{\lambda}}, \cdot) = 0$ in the variables $\boldsymbol{\beta}$ and $\boldsymbol{\rho}$ while fixing $\hat{\boldsymbol{\lambda}}$, which allows us to estimate the model parameters $\hat{\boldsymbol{\beta}}$ and $\hat{\boldsymbol{\rho}}$. In particular, Figure 6 supports this statement where we see that solving $\psi(\cdot, \hat{\boldsymbol{\lambda}}, \cdot) = 0$ allows us to estimate $\beta_1$ and $\beta_2$ with reasonably low variance. Moreover, an important aspect of parameter estimation is proving its consistency. Namely, demonstrating a Central Limit Theorem (CLT) result that shows the concentration of $\hat{\boldsymbol{\beta}}$ around the true $\boldsymbol{\beta}$ as well as for $\hat{\boldsymbol{\rho}}$. We currently support this statement through simulations as depicted in Figures 6 and 7. Note however that, given the concentration of $\hat{\boldsymbol{\lambda}}$, we believe that such consistency can be ensured with additional assumptions on the function $\psi$ in equation 50 and in particular the existence and uniqueness of solution to the equation $\psi(\cdot, \hat{\boldsymbol{\lambda}}, \cdot) = 0$.

$$\psi : (\boldsymbol{\beta}, \boldsymbol{\lambda}, \boldsymbol{\rho}) \mapsto \begin{pmatrix} f_r(\lambda_1) - \sum_{i=1}^2 \beta_i \rho_{1i}^3 \\ h_r(\lambda_1)\rho_{11} - \sum_{i=1}^2 \beta_i \alpha_{i1} \rho_{1i}^2 \\ h_r(\lambda_1)\rho_{12} - \sum_{i=1}^2 \beta_i \alpha_{i2} \rho_{1i}^2 \\ f_r(\lambda_2) - \sum_{i=1}^2 \beta_i \theta_{2i} \rho_{2i}^2 \\ h_r(\lambda_2)\theta_{21} - \frac{\eta^2}{3} r(\lambda_1)\rho_{11} - \sum_{i=1}^2 \beta_i \alpha_{i1} \rho_{2i}^2 + \rho_{11} \sum_{i=1}^2 \beta_i \rho_{1i} \rho_{2i}^2 \\ h_r(\lambda_2)\theta_{22} - \frac{\eta^2}{3} r(\lambda_1)\rho_{12} - \sum_{i=1}^2 \beta_i \alpha_{i2} \rho_{2i}^2 + \rho_{12} \sum_{i=1}^2 \beta_i \rho_{1i} \rho_{2i}^2 \\ h_r(\lambda_2)\rho_{21} - \sum_{i=1}^2 \beta_i \theta_{2i} \rho_{2i} \alpha_{i1} \\ h_r(\lambda_2)\rho_{22} - \sum_{i=1}^2 \beta_i \theta_{2i} \rho_{2i} \alpha_{i2} \\ \left[\lambda_2 + \frac{2}{3} r(\lambda_2)\right] \eta - \sum_{i=1}^2 \beta_i \theta_{2i} \rho_{1i} \rho_{2i} \end{pmatrix} \tag{50}$$

**Case $\gamma = 1$:** As we discussed earlier, in the case $\gamma = 1$ the limiting spectral measure $\nu$ becomes equal to the semi-circle law $\mu$ described in the first deflation step. Moreover, the system of equations in equation 13 reduces to the following equations, for $j \in [2]$, which will be needed subsequently.

$$\begin{cases} f_r(\lambda_2) = \sum_{i=1}^2 \beta_i \theta_{2i} \rho_{2i}^2, \quad h_r(\lambda_2)\theta_{2j} - \frac{\eta^2}{3} r(\lambda_1)\rho_{1j} = \sum_{i=1}^2 \beta_i \alpha_{ij} \rho_{2i}^2 - \rho_{1j} \sum_{i=1}^2 \beta_i \rho_{1i} \rho_{2i}^2 \\ h_r(\lambda_2)\rho_{2j} = \sum_{i=1}^2 \beta_i \theta_{2i} \rho_{2i} \alpha_{ij}, \quad \left[\lambda_2 + \frac{2}{3} r(\lambda_2)\right] \eta = \sum_{i=1}^2 \beta_i \theta_{2i} \rho_{1i} \rho_{2i} \end{cases} \tag{51}$$

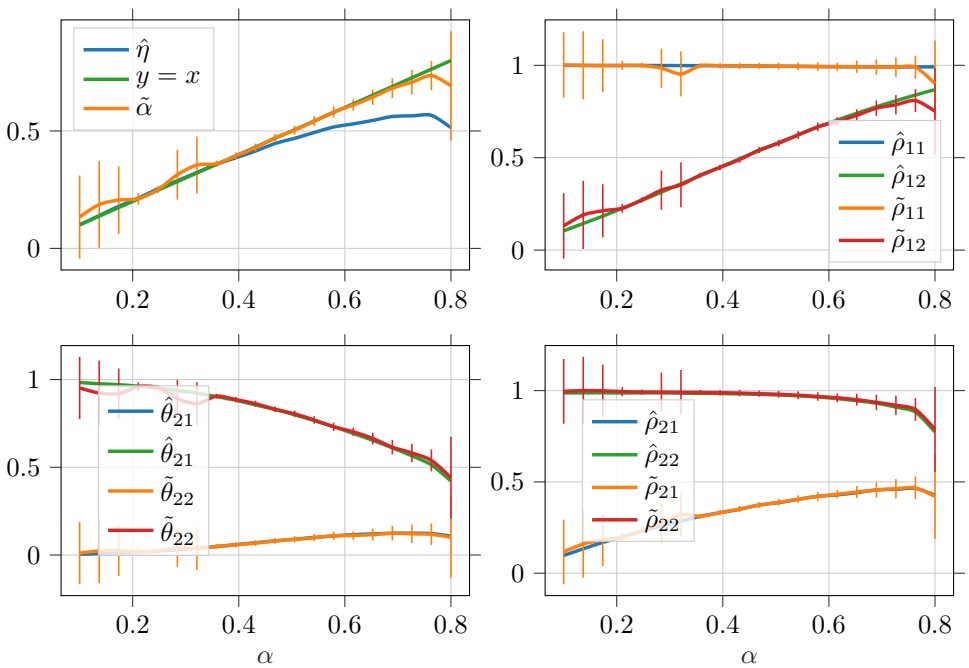

Figure 7: Estimation of **Alignments** as described in Section 3 from one realization of the random tensor $\mathcal{T}_1$. We considered $\beta_1 = 15$, $\beta_2 = 5$, $\gamma = 1$, $p = 100$ while varying $\alpha$. The curves are averaged over 100 realizations of $\mathcal{T}_1$. The hats correspond to simulations while tildes correspond to the estimated alignments as per Section 3.

### E.3 SIMULATED AND ASYMPTOTIC ALIGNMENTS AT FIRST DEFLATION STEP

We observe a near perfect estimation of all the problem parameters of the first deflation step of $\gamma$-orthogonalized tensor deflation in Figure 8.

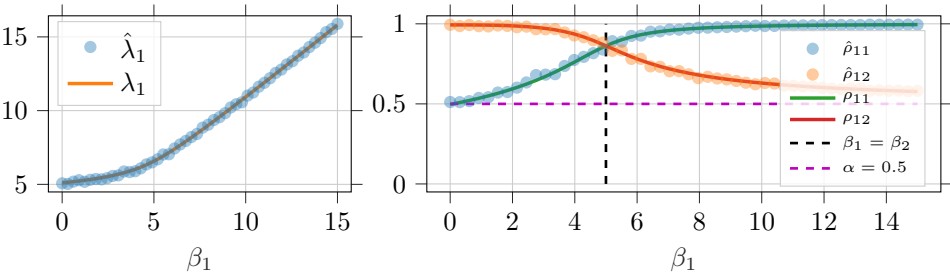

Figure 8: **Simulated** vs **theoretical** asymptotic singular value and alignments corresponding to the first deflation step as per Theorem 2. We considered $\beta_2 = 5$, $\alpha = 0.5$, $p = 100$ and varying $\beta_1 \in [0, 15]$. The system of equations in equation 8 is solved numerically and initialized with the simulated singular value and alignments (dotted curves) from one realization of $\mathcal{T}_1$.

### E.4 CONCAVITY OF ALIGNMENTS IN $\gamma$

An important property of interest is the behavior of the asymptotic alignments as a function of $\gamma$, we show that empirically in Figures 7 and 10 for different stages of $\gamma$-orthogonalized tensor deflation.

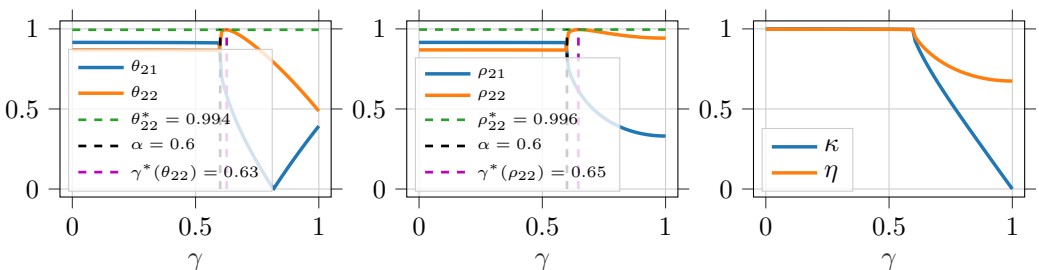

Figure 9: Asymptotic alignments of the second deflation varying the hyper-parameter $\gamma$. We considered $\beta_1 = 10$, $\beta_2 = 8$ and $\alpha = 0.6$.

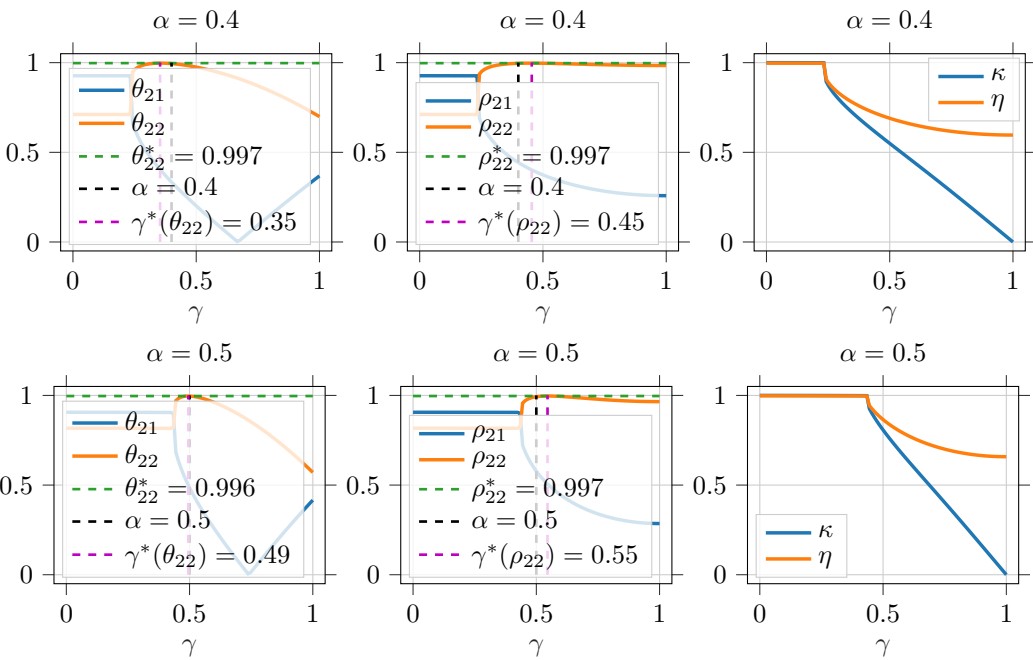

Figure 10: Asymptotic alignments of the second deflation step in terms of $\gamma$ and $\alpha$. We considered $\beta_1 = 10$ and $\beta_2 = 8$.

## F  ALGORITHMS

Algorithm 1 below, implements the fixed point equation in Definition 3.3 which allows the computation of the Stieltjes transform at the second deflation step.

---

**Algorithm 1** Stieltjes Transform by Fixed Point

**Input:** $z \in \mathbb{C} \setminus \mathrm{supp}(\nu)$ and $\tau$.
- Initialize $a$ and $b$.
**while** no convergence **do**
    - Update $a \leftarrow \frac{-1}{3(2b+z)}$.
    - Update $b \leftarrow \frac{-1}{3(a+z-\tau b)}$.
**end while**
**Output:** $a, b$ and Stieltjes transform $q = a + 2b$.

---

Algorithm 2 implements our $\gamma$-orthogonalized tensor deflation procedure which was carefully constructed in Section 3.1.

---

**Algorithm 2** $\gamma$-Orthogonalized Tensor Deflation Algorithm

---

**Input:** Tensor $\mathcal{T} \in \mathbb{R}^{p \times p \times p}$ and step size $\epsilon \in [0, 1]$.

# Perform orthogonalized deflation:

1- Compute $\hat{\lambda}_1 \hat{u}_1 \otimes \hat{v}_1 \otimes \hat{w}_1$ as best rank-1 approximation of $\mathcal{T}$.

2- Compute $\hat{\lambda}_2 \hat{u}_2 \otimes \hat{v}_2 \otimes \hat{w}_2$ as best rank-1 approximation of $\mathcal{T} \times_1 (\boldsymbol{I}_p - \gamma \hat{u}_1 \hat{u}_1^\top)$ for $\gamma = 1$.

# Estimate underlying model parameters:

3- Compute $\hat{\eta} \leftarrow |\langle \hat{v}_1, \hat{v}_2 \rangle|$.

4- Estimate $\hat{\boldsymbol{\beta}} = (\hat{\beta}_1, \hat{\beta}_2, \hat{\alpha})$ and $\hat{\boldsymbol{\rho}} = (\hat{\rho}_{1i}, \hat{\rho}_{2i}, \hat{\theta}_{2i} \mid i \in [2])$ by fixing $\hat{\boldsymbol{\lambda}} = (\hat{\lambda}_1, \hat{\lambda}_2, \hat{\eta})$ and solving $\psi(\hat{\boldsymbol{\beta}}, \hat{\boldsymbol{\lambda}}, \hat{\boldsymbol{\rho}}) = 0$ in $\hat{\boldsymbol{\beta}}$ and $\hat{\boldsymbol{\rho}}$ with $\psi$ defined in equation 50.

# Estimate optimal $\gamma$:

5- Initialize $\gamma = 1$ and $\hat{\kappa} = 10^{-5}$.

6- Initialize two empty lists $L_\gamma$ and $L_\rho$.

**while** The maximum in $L_\rho$ is not reached **do**

    7- Set $x_0 \leftarrow (\hat{\lambda}_2, \hat{\kappa}, \hat{\eta}, \hat{\theta}_{2i}, \hat{\rho}_{2i} \mid i \in [2])$.

    8- Estimate $(\hat{\lambda}_2, \hat{\kappa}, \hat{\eta}, \hat{\theta}_{2i}, \hat{\rho}_{2i} \mid i \in [2])$ by solving the system in equation 13 initialized with $x_0$ and for $(\beta_1, \beta_2, \alpha) = (\hat{\beta}_1, \hat{\beta}_2, \hat{\alpha})$ and $\gamma$.

    9- Append $L_\gamma$ with $\gamma$.

    10- Append $L_\rho$ with $\max\{\hat{\rho}_{21}, \hat{\rho}_{22}\}$.

    11- Update $\gamma \leftarrow \gamma - \epsilon$.

**end while**

12- Set optimal $\gamma$ as $\gamma^* \leftarrow L_\gamma[\arg\max(L_\rho)]$.

# Perform orthogonalized deflation with $\gamma^*$:

13- Compute $\hat{\lambda}_2 \hat{u}_2 \otimes \hat{v}_2^* \otimes \hat{w}_2^*$ as best rank-1 approximation of $\mathcal{T} \times_1 (\boldsymbol{I}_p - \gamma^* \hat{u}_1 \hat{u}_1^\top)$.

14- Compute $\hat{\lambda}_2 \hat{u}_2^* \otimes \hat{v}_2 \otimes \hat{w}_2^*$ as best rank-1 approximation of $\mathcal{T} \times_2 (\boldsymbol{I}_p - \gamma^* \hat{v}_1 \hat{v}_1^\top)$.

# Re-estimate the first component by simple deflation:

15- Compute $\hat{\lambda}_1 \hat{u}_1^* \otimes \hat{v}_1^* \otimes \hat{w}_1^*$ as best rank-1 approximation of $\mathcal{T} - \min\{\hat{\beta}_1, \hat{\beta}_2\} \hat{u}_2^* \otimes \hat{v}_2^* \otimes \hat{w}_2^*$.

**Output:** Estimates of the signal components $(\max\{\hat{\beta}_1, \hat{\beta}_2\}, \hat{u}_1^*, \hat{v}_1^*, \hat{w}_1^*), (\min\{\hat{\beta}_1, \hat{\beta}_2\}, \hat{u}_2^*, \hat{v}_2^*, \hat{w}_2^*)$.

---

# G   ADDRESSING LIMITATIONS & FUTURE WORK

We have showcased a concrete example where RMT/RTT allows us to understand and even improve signal recovery from low-rank asymmetric spiked random tensors. To the best of our knowledge, this is the **first time** where an asymptotic characterization of the considered deflation method is carried out. For the sake of clarity, we limited our detailed analysis to the more intuitive rank-2 order-3 model. On a further note, we outline that our actual results already pave a new path towards the analysis and improvement of more sophisticated tensor methods and models, by means of random tensor theory, thereby impacting tensor-based machine learning methods and many other applications. In the following, we list some extensions to address some limitations in our current work.

- **Extension to Higher Ranks & Orders.** We present the meta steps that allow to obtain our results and generalize beyond that to low rank tensor models of arbitrary ranks and/or orders. The machinery behind the generalized deflation step theoretical analysis framework (3) lies in the application of Stein's lemma (principally, among others) to estimate the different problem parameters, alignments in particular. Given that this is a non-amortized approach, the number of model parameters and alignments grows exponentially in the number of deflation steps, which makes the analysis very tedious by hand. Upon the use of symbolic solvers able to sequentially apply lemmas like that of Stein, the generalization of our to higher orders/ranks becomes fairly straightforward. To give the reader an idea of how that would look like, we refer them to Seddik et al. (2023). In particular, Theorem 3.4 outlines a system of equations describing the Hotelling-type asymmetric tensor deflation for general ranks/orders.

- **Evaluation on Real-World Data.** While the application in real-world contexts is quite straightforward, given the plethora of applications listed in the introduction, we limit this work to showcase the potential of utilizing a theoretical algorithmic interplay to design new theoretically interpretable, robust and highly accurate tensor deflation/decomposition methods. We benchmark against prevalent tensor decomposition approaches CP and Tucker,

and show that our approach performs optimally while the latter collapse in the presence of higher levels of correlation.

- **On a Theoretical Level.** We highlight the following points

  - Our main results rely on Assumptions 3.1 and 3.4 which basically suppose the almost convergence of the singular values and alignments of interest. Similar Assumptions were made and discussed (in (Goulart et al., 2021; Seddik et al., 2021)) which also relied on a RMT approach. A formal proof of these statements is still required and would make our analysis more complete.

  - The second point concerns the existence and uniqueness of the solutions of the involved systems of equations. In essence, the asymptotic behavior of the second deflation step is described by a set of seven polynomial equations in $\lambda_2$ and the alignments $\theta_{2i}, \rho_{2i}, \kappa$ and $\eta$. Again, we do not address the existence and uniqueness of such solutions, and we solve the system in equation 13 numerically starting from the simulated singular value and alignments from one realization of $\mathcal{T}_1$. Contrarily to the first deflation step, we highlight that the Stieltjes transform $q(z)$ depends on the alignment $\kappa$. Therefore, we alternate solving the system in equation 13 with the fixed point equations in equation 11 for $\tau = \gamma\kappa^2 - 1 + \kappa(\gamma - 1)$ as per Theorem 3.

  - The third point concerns a proof of consistency of the underlying model parameters estimation. Specifically, proving a Central Limit Theorem (CLT) about the convergence of our estimates to the true parameters and the related conditions.

  - On a further note, we also outline that there might also exist a theoretical-algorithmic spectral gap, that needs to be determined for the present deflation procedure, in the same spirit as that in Richard & Montanari (2014) for the rank-1 case.

We do not address these questions in our present analysis and we defer them to a future study.

