# OpenReview forum: "$\gamma$-Orthogonalized Tensor Deflation: Towards Robust \& Interpretable Tensor Decomposition in the Presence of Correlated Components"
_ICLR.cc/2024/Conference — Submitted to ICLR 2024_

### Official Review · Reviewer_Een1 · 2023-11-01

**Soundness:** 2 fair
**Presentation:** 1 poor
**Contribution:** 2 fair
**Rating:** 3
**Confidence:** 3

**Summary:**

The authors consider the problem of finding the components of the expansion of a tensor with noise into a sum of rank-one tensors. In fact, only the case of rank-2 order-3 model is considered. Emphasis is placed on the so-called non-orthogonal case when the components are correlated. For this case, the paper propose an algorithm containing the parameter $\gamma$ for the sequential construction of the desired decomposition. This parameter $\gamma$ is chosen adaptively by repeatedly running the algorithm. Asymptotics of the decomposition parameters are given as a theoretical analysis. A small number of numerical examples on synthetic data are given.

**Strengths:**

- Robust algorithm of tensor decomposition is build and analyzed for narrow specific case (rank-2 order-3 CP tensor with noise).

**Weaknesses:**

- The writing style leaves a lot to be desired as the text is quite hard to read. The narrative is non-linear, many important things for understanding are put in the Appendix, even the notations. Section 3.2 MAIN ALGORITHM SKETCH, where the essence of the underlying algorithm is revealed, is overflowing with links to Appendix.

- The paper is widely cited following paper (Seddik et al, 2021), so it is difficult to understand what the current paper is about without reading the cited paper.

- The most important weakness, in my opinion, is too few experiments that have been conducted only on synthetic data. In general, experimental section 4 is very short, subsections 4.2 and 4.3 consist of only figures. Thus, it is not clear at all the relevance of this paper, where practically the presented results can be applied.

- Neither is there any comparison with other existing methods.

- Only rank-2 order-3 model is considered, which significantly reduces the breadth of application of the method and, as a consequence, its practical value.

**Questions:**

- Weighted sum of 1-rank tensors is called CANDECOMP/PARAFAC tensor Decomposition (CP) and there is quite a large theoretical basis for it. Have you used or compared with other existing approaches for approximate tensor construction in CP?

- Can your work be extended to other tensor decompositions such as Tensor Train (TT), Tucker and others?

---

> ### Author Response · Authors · 2023-11-22
>
> We thank the reviewer very much for their valuable feedback. We address their questions and comments as follows:
>
> **Outline**. We have significantly revisited our manuscript, as outlined in our general comment.
> - The main algorithm is motivated and presented in details in Section 3.4, which follows a thorough and complete theoretical construction of our algorithm. We list the main assumptions and theorems, outline their value within our framework and visualize them with adequate simulations.
> - (Seddik et al, 2021) lays the ground to a new angle to approach spectral tensor decomposition problems. Our paper follows the same path to construct our algorithm. While the reader is referred to check that paper for further details about random tensor theory based spectral tensor decomposition (proof of Theorem 1 in Section 3.2.1 of our paper for instance), our paper outlines the detailed construction of all our results in Section 3 of the main paper and provides a self-contained review of concepts of random matrix/tensor theory (RMT/RTT) in Appendix B to guide the unfamiliar reader.
>
> **Experiments.** We update our experiments section, as outlined in our general comment. This further addresses the reviewer's specific questions. We list the relevant section again here for their convenience:
>
> - We now focus our empirical evaluation on two representative experiments and provide detailed explanations of both. In particular, we outline how our algorithmic approach performs compared to state-of-the-art and how our developed theory matches our predictions in practice with extensive experiments. In addition, we have also added separate visualizations for our theorems and assumptions throughout Section 3 and provided a detailed set of supplementary experiments in Appendix E.
>
> **Application.** We agree with the reviewer that the rank-2 order-3 model limits the breadth of applications of the considered model in general. However, this work only presents the rank-2 order-3 model for its accessibility and to show the potential of the considered analysis. For higher ranks/orders, a similar construction can be done (hints on that in Appendix G), which was previously developed for a similar model- Hotelling-type tensor deflation (See Theorem 3.4 in [1])
>
> **Questions.**
> - We thank the reviewer for their great question! We have further benchmarked our proposed algorithm against prevalent spectral tensor decomposition methods that are widely used in practice, CANDECOMP/PARAFAC (CP) and Tucker. We show that the latter collapse in the presence of higher levels of correlation $\alpha$ and fail to recover the signal of interest. We outline the details of that in Figure 5 and its subsequent explanation in Section 4/Benchmarks. Initially, we have not considered this benchmark highly relevant, as we only considered spectral decomposition algorithms.
> - Our work, for now, only considers spectral decomposition algorithms given the rich theoretical basis in RMT/RTT. Tensor Train (TT) and Tucker are, however, compression-based decomposition algorithms.
>
> We invite the reviewer's feedback and suggestions, which we will gladly consider to further improve our paper.
>
> [1]: Seddik, M. E. A., Guillaud, M., Decurninge, A., & Goulart, J. H. D. M. (2023). On the Accuracy of Hotelling-Type Asymmetric Tensor Deflation: A Random Tensor Analysis. arXiv preprint arXiv:2310.18717.

---

### Official Review · Reviewer_rV4h · 2023-11-01

**Soundness:** 3 good
**Presentation:** 1 poor
**Contribution:** 3 good
**Rating:** 5
**Confidence:** 3

**Summary:**

This paper studies the tensor decomposition problem on the spiked tensor model. A novel $\gamma$-Orthogonalized Tensor Deflation that extends from the standard deflation is proposed. A random tensor theory analysis has been established for the proposed algorithm on a rank-2 case. Numerical results are aligned with the theory.

**Strengths:**

The paper has solid analytic results, and the proof seems overall correct (I didn't check too carefully). The numerical results on synthetic data seem convictive.

**Weaknesses:**

The paper is poorly presented, messily organized, and hard to follow. It seems the authors just put everything together at the last minute. The writing style of each paper component is not consistent. A complete polish must be done before the paper can be published.

Experiments on real-world applications are needed to further establish the empirical performance of the proposed method.

**Questions:**

See weakness

---

> ### Author Response · Authors · 2023-11-22
>
> We thank the reviewer for their comment. We address their concerns as follows
>
> **Outline.** Indeed, it was a real challenge for us to present our work in such a way that outlines the motivation behind our approach, illustrates the main steps for developing the theory and constructing the algorithm, and at the same time extensively evaluate our claims empirically. In this spirit, we have substantially revisited our manuscript and list the main changes in a general comment.
>
> **Experiments** We have further benchmarked our proposed algorithm against prevalent spectral tensor decomposition methods that are widely used in practice, CANDECOMP/PARAFAC (CP) and Tucker. We show that the latter collapse in the presence of higher levels of correlation and fail to recover the signal of interest. We outline the details of that in Figure 5 and its subsequent explanation in Section 4/Benchmarks. Regarding evaluation on real-world data, we defer that to subsequent work. Yet, we would like to outline that our model encompasses a rich class of problems, as outlined in the Introduction Section.
>
> We invite the reviewer's feedback and suggestions, which we will gladly consider to further improve our paper.

---

### Official Review · Reviewer_54VG · 2023-11-05

**Soundness:** 4 excellent
**Presentation:** 3 good
**Contribution:** 3 good
**Rating:** 8
**Confidence:** 5

**Summary:**

The paper deals with low rank tensor recovery under the spike model. The paper particularly focuses on a scheme named "deflation" which calculates leading singular value / vectors one by one and deflates the original tensor, i.e. subtracts the outer product of the calculated top vector times singular value. In a matrix scenario (order = 2) this method is rather straightforward, because singular vectors are orthogonal. In tensors (order >=3) orthogonality is not always satisfied. This makes deflation more challenging. This is the topic the paper addresses. The paper treats the case order=3, rank=2 in details and outlines the schema for higher order and ranks.

**Strengths:**

Paper addresses a very interesting and challenging question. While a series of papers have studies the rank-1 (PCA) problem for tensors, the higher rank estimation problem comes with its own challenges, mostly due to orthogonality and the deflation scheme that authors discuss here.
Authors provide an algorithm which is backed by random tensor theory. Authors also provide numerical simulations that prove that the theoretical asymptotics are alined with empirical (finite n) values.
Authors outline how to generalize their results to more complex scenarios.
The works presentation is self-contained and well referenced, which makes it accessible to a wide audience, despite the topic's technicality.

**Weaknesses:**

Given how cumbersome notations and formalism get when working with tensors, it always helps to make simplifying assumptions to prove your point in the smallest yet representative case. Authors have done a pretty good job at it here, while I could still imagine simpler problem formulations. Two suggestions / questions below.

**Questions:**

1. Why did you not consider a (simpler) symmetric case where u=v=w for simplifying notations and exposing the main results more easily?
2. Did you try to express the optimization problem in gamma as an objective function minimization problem? Can the problem benefit a joint optimization in both singular value / vectors and gamma?

---

> ### Author Response · Authors · 2023-11-22
>
> We thank the reviewer very much for their positive feedback and valuable suggestions.
>
> We answer the questions as follows:
>
> - For the symmetric case where $u=v=w$, this indeed seems a simpler case to consider. However, when considering such assumption, we impose further constraints on the structure of the noise tensor $\mathcal{W}$. The noise entries $\mathcal{W}_{i,j,k}$ are not i.i.d anymore and further steps would be needed to estimate the variance given a certain entry.
> - We thank the reviewer for bringing this point to our attention. Indeed, at the second deflation step of the $\gamma$-orthogonalized tensor deflation algorithm (in Appendix F, Algorithm 2), we are jointly optimizing $\gamma$ and the alignments. Furthermore, we empirically show that the different alignments are concave functions of $\gamma$ in Appendix E.4 and emphasize that in Section 3.4, which further explains the importance of our approach and provides theoretical guarantees on the quality of our recovery given $\gamma^*$.

---

### Official Review · Reviewer_fjzE · 2023-11-10

**Soundness:** 3 good
**Presentation:** 2 fair
**Contribution:** 2 fair
**Rating:** 5
**Confidence:** 3

**Summary:**

This paper addresses the challenge of recovering a low-rank tensor signal with potentially correlated components from a random noisy tensor, which is known as the spiked tensor model. The authors propose a solution for the non-orthogonal case using a parameterized deflation procedure, referred to as γ-orthogonalized tensor deflation. Then an efficient tensor deflation algorithm is introduced, which optimizes the parameter used in the deflation mechanism. In addition, this paper provides a detailed theoretical analysis for the case of rank-2 order-3 tensor model and suggests a general structure for handling the problem for arbitrary rank/order tensors. These findings aim to have a broader impact in machine learning and beyond.

**Strengths:**

1. The article uses the newly proposed random tensor theory and tensor deflation method to solve the spiked tensor recovery problem when the signal components are $\gamma$-orthogonal. Basically, the proposed method has a wider range of practicality.
2. A series of numerical experiments are conducted to verify the applicability of the proposed method in different situations.

**Weaknesses:**

1. According to the presented analysis, expanding this method to rank-$r$ and higher-order situations seems to be very cumbersome. Although the authors state that symbolic solvers can be used to solve such higher-order and higher-rank problems, it could be beneficial to provide some more details for the readers to follow.
2. There are many figures in the text, but there are few related explanations. More explanations should be added to make such figures to be more meaningful.
3. The authors put too much stuff in the supplementary materials, which is not conducive for the readers' reading. For example, the proposed algorithms should be placed in the main text.
4. The introduction of the numerical experiments should be more detailed, such as how the simulated data is generated.

**Questions:**

See the weaknesses above.

---

> ### Author Response · Authors · 2023-11-22
>
> We thank the reviewer very much for their valuable feedback and suggestions, which has opened many avenues for the improvement of our work. We have substantially revisited our paper and we list the most important changes as a general comment.
>
> To address the reviewer's specific questions:
>
> 1. **Extension to Higher-Rank and Higher-Order Problems.** In addition to the Generalized Deflation Step Procedure in Section 3.1, we now motivate every step in our developed theory/algorithm in detail in the main text. We have considered the extension to higher ranks and/or orders in more detail in Appendix G, where we also list some of the key lemmas to producing our results. In addition, we point the reader to Theorem 3.4 in [1], where the extension follows a similar approach to ours and is done in full and in a high level of detail for the Hotelling-type deflation.
> 2.  **Experiments Section.** We focused our experimental evaluation on 2 representative experiments and provide detailed explanations of both. In particular, we outline how our algorithmic approach performs compared to state-of-the-art and how our developed theory matches our predictions in practice with extensive experiments. In addition, we have also added separate visualizations for our theorems and assumptions throughout Section 3 and provided a detailed set of supplementary experiments in Appendix E.
> 3. **Outline of the Paper.** We substantially revisited our manuscript, as outlined as part of our general comment. The proposed algorithm is constructed and explained in a high level of detail in Section 3.
> 4. **Experimental Setup.** We provided more details of our experimental setup in Section 4 (Experimental Setup). In addition, an easy-to-follow code sample is provided in Appendix E.1 to show exactly how our data is generated. A code repository will also be provided to ensure reproducibility.
>
> We welcome the reviewer's additional questions and/or suggestions!
>
> [1]: Seddik, M. E. A., Guillaud, M., Decurninge, A., & Goulart, J. H. D. M. (2023). On the Accuracy of Hotelling-Type Asymmetric Tensor Deflation: A Random Tensor Analysis. arXiv preprint arXiv:2310.18717.

---

### Author Response · Authors · 2023-11-22
**Manuscript Revision**

We thank the reviewers very much for their valuable feedback and suggestions. Following the latter, we have substantially revisited our manuscript.

We list the main changes as follows:

**Outline \& Theory: Section 3.** We fully restructured the main section of our paper. We provide the full detailed construction of our theory and subsequently our proposed algorithm. We emphasize the motivation behind each step and visualize the different assumptions and results with representative simulations.

**Empirical Evaluation.**

- We now focus our empirical evaluation on two representative experiments and provide detailed explanations of both. In particular, we outline how our algorithmic approach performs compared to state-of-the-art and how our developed theory matches our predictions in practice with extensive experiments. In addition, we have also added separate visualizations for our theorems and assumptions throughout Section 3 and provided a detailed set of supplementary experiments in Appendix E.

- We have further benchmarked our proposed algorithm against prevalent spectral tensor decomposition methods that are widely used in practice, CANDECOMP/PARAFAC (CP) and Tucker. We show that the latter collapse in the presence of higher levels of correlation $\alpha$ and fail to recover the signal of interest. We outline the details of that in Figure 5 and its subsequent explanation in Section 4/Benchmarks.

**Extensions.** We defer the Limitations \& Future Work component to Appendix G and provide some more detail about the possible extensions to our work, both theoretical and empirical.

---

### Meta-Review · Area_Chair_XwiA · 2023-12-05

**Metareview:**

This paper considers the problem of recovering a low-rank tensor from a random noisy tensor. The paper proposes a new deflation procedure which can lead to a new tensor deflation algorithm when the components are not orthogonal. The paper mostly focused on the analysis on rank-2 order-3 tensors. The main concerns in the reviews are the clarify of presentation, and the limitation of the algorithm to rank-2 order-3 tensors (extending the algorithm seems cumbersome).

**Justification For Why Not Higher Score:**

The paper is on the borderline. I feel the approach is interesting but as some reviewers noted it's not clear how this compare with some existing approach, and it's difficult to generalize.

**Justification For Why Not Lower Score:**

N/A

---

### Decision · Program_Chairs · 2024-01-16

Reject